# IL-SOAR : Imitation Learning with Soft Optimistic Actor cRitic

Stefano Viel [* 1]   Luca Viano [* 1]   Volkan Cevher [1]

## Abstract

This paper introduces the SOAR framework for imitation learning. SOAR is an algorithmic template that learns a policy from expert demonstrations with a primal dual style algorithm that alternates cost and policy updates. Within the policy updates, the SOAR framework uses an actor critic method with multiple critics to estimate the critic uncertainty and build an optimistic critic fundamental to drive exploration.

When instantiated in the tabular setting, we get a provable algorithm with guarantees that matches the best known results in the desired accuracy parameter $\epsilon$.

Practically, the SOAR template can boost the performance of *any* imitation learning algorithm based on Soft Actor Critic (SAC). As an example, we show that SOAR can boost consistently the performance of the following SAC-based imitation learning algorithms: $f$-IRL, ML-IRL and CSIL. Overall, thanks to SOAR, the required number of episodes to achieve the same performance is reduced by half.[1]

## 1. Introduction

Several recent state of the art imitation learning (IL) algorithms (Ni et al., 2021; Zeng et al., 2022; Garg et al., 2021; Watson et al., 2023; Viano et al., 2022b) are built on Soft Actor Critic (SAC) (Haarnoja et al., 2018) to perform the policy updates. SAC uses *entropy* regularized policy updates to maintain a strictly positive probability of taking each action. However, this is known to be an inefficient exploration strategy if deployed alone (Cesa-Bianchi et al., 2017).

*Equal contribution   [1]EPFL, Lausanne.   Correspondence to: Stefano Viel <stefano.viel@epfl.ch>, Luca Viano <luca.viano@epfl.ch>.

*Proceedings of the $42^{nd}$ International Conference on Machine Learning*, Vancouver, Canada. PMLR 267, 2025. Copyright 2025 by the author(s).

[1]Project code available at https://github.com/stefanoviel/SOAR-IL/tree/master

Indeed, several recent theoretical imitation learning achieve performance guarantees by adding exploration bonuses on top of the regularized policy updates, which encourage the learner to visit state-action pairs that have not been visited previously. Unfortunately, such works are only available in the tabular setting (Shani et al., 2021; Xu et al., 2023) and in the linear setting (Viano et al., 2024). The design of the exploration bonuses in these works is strictly tight to the tabular or linear structure of the transition dynamics, therefore, these analyses offer little insight on how to design an efficient exploration mechanism using neural network function approximation.

There is, therefore, a lack of a technique that satisfies the following two requirements.

- It is statistically and computationally efficient in the tabular setting.
- It can be implemented easily in continuous states and actions problems requiring neural networks function approximation.

In this paper, we present a general template, dubbed Soft Optimistic Actor cRitic Imitation Learning (SOAR-IL) satisfying these requirements.

The main idea is to act according to an *optimistic* critic within the SAC block on which many IL algorithms rely. Here, optimism means appropriately underestimating the expected cumulative cost incurred by playing a policy in the environment. This principle known as *optimism in the face of uncertainty* has led to several successful algorithms in the bandits community.

While optimism is often achieved using the structure of the problem (tabular, linear, etc.), in this work, we build optimistic estimators using an ensemble technique. That is, multiple estimators for the same quantity are maintained and aggregated to obtain an optimistic estimator. This technique scales well with deep imitation learning. To summarize, we have the following contributions.

**Theoretical contribution**   We show that there exists a computationally efficient algorithm that uses an ensemble based exploration technique that gives access to $\mathcal{O}(\epsilon^{-2})$ expert trajectories and $\mathcal{O}(\epsilon^{-2})$ interactions in a tabular MDP outputs a policy such that its cumulative expected cost is at

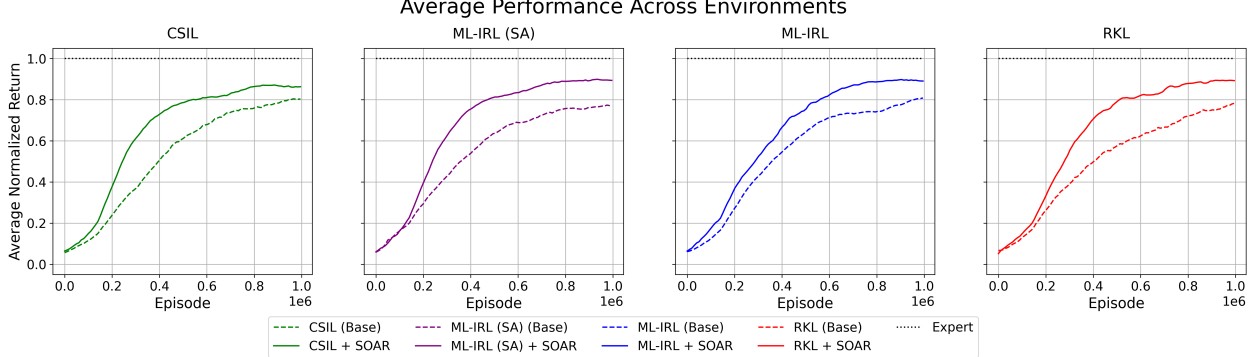

Figure 1: Summary of experimental results. Each plot compares the average normalized return across 4 MuJoCo environments with 16 expert trajectories for a base algorithm and its SOAR-enhanced version. SOAR replaces the single critic in SAC-based methods with multiple critics to compute an optimistic estimate. Across all algorithms, incorporating SOAR consistently improves performance. ML-IRL (SA) stands for ML-IRL (Zeng et al., 2022) from expert state-action demonstrations.

most $\epsilon$ higher than the expert cumulative expected cost with high probability.

**Practical Contribution** We apply an ensemble-based exploration technique, SOAR, to boost the performance of deep imitation learning algorithms built on SAC, demonstrating its effectiveness on MuJoCo environments. Specifically, we show that incorporating SOAR consistently boosts the performance of base methods such as Coherent Soft Imitation Learning (CSIL)(Watson et al., 2023), Maximum Likelihood IRL (ML-IRL)(Zeng et al., 2022) and RKL (Ni et al., 2021). As shown in Figure 1, our approach consistently outperforms the base algorithms across all MuJoCo environments. Notably, SOAR achieves the best performance of the baselines requiring only approximately half the number of learning episodes.

## 2. Preliminaries and Notation

The environment is abstracted as Markov Decision Process (MDP) (Puterman, 1994) which consists of a tuple $(\mathcal{S}, \mathcal{A}, P, c, \boldsymbol{\nu}_0, \gamma)$ where $\mathcal{S}$ is the state space, $\mathcal{A}$ is the action space, $P : \mathcal{S} \times \mathcal{A} \to \Delta_{\mathcal{S}}$ is the transition kernel, that is, $P(s'|s, a)$ denotes the probability of landing in state $s'$ after choosing action $a$ in state $s$. Moreover, $\boldsymbol{\nu}_0$ is a distribution over states from which the initial state is sampled. Finally, $c : \mathcal{S} \times \mathcal{A} \to [0, 1]$ is the cost function, and $\gamma \in [0, 1)$ is called the discount factor.

**Value functions and occupancy measures** We define the state value function at state $s \in \mathcal{S}$ for the policy $\pi$ under the cost function $c$ as $V_c^\pi(s) \triangleq \mathbb{E}\left[\sum_{h=0}^\infty \gamma^h c(s_h, a_h)|s_1 = s\right]$. The expectation over both the randomness of the transition dynamics and the one of the learner's policy. Another convenient quantity is the occupancy measure of a policy $\pi$ denoted

as $d^\pi \in \Delta_{\mathcal{S} \times \mathcal{A}}$ and defined as follows $d^\pi(s, a) \triangleq (1 - \gamma) \sum_{h=0}^\infty \gamma^h \mathbb{P}[s, a \text{ is visited after } h \text{ steps acting with } \pi]$. We can also define the state occupancy measure as $d^\pi(s) \triangleq (1 - \gamma) \sum_{h=0}^\infty \gamma^h \mathbb{P}[s \text{ is visited after } h \text{ steps acting with } \pi]$.

**Imitation Learning** In imitation learning, the learner is given a dataset $\mathcal{D}_{\pi_E}$ of expert trajectories collected by an unknown expert policy $\pi_E$.[2] By trajectory $\boldsymbol{\tau}^k$, we mean the sequence of states and actions sampled rolling out the policy $\pi_k$ for a number of steps sampled from the distribution $\text{Geometric}(1 - \gamma)$. Given $\mathcal{D}_{\pi_E}$, the learner adopts an algorithm $\mathcal{A}$ to learn a policy $\pi^{\text{out}}$ such that $\left\langle \boldsymbol{\nu}_0, V_{c_{\text{true}}}^{\widehat{\pi}^k} - V_{c_{\text{true}}}^{\pi_E} \right\rangle \leq \epsilon$ with high probability.

We use the notation $\mathbf{e}_s$ to denote a vector in $\mathbb{R}^{|\mathcal{S}|}$ zero everywhere but in the coordinate corresponding to the state $s$ ( for an arbitrary ordering of the states). Analogously, we use $\mathbf{e}_{s,a}$ to denote a vector in $\mathbb{R}^{|\mathcal{S}||\mathcal{A}|}$ zero everywhere but in the $(s, a)^{th}$ entry which equals one.

## 3. The Algorithm

In this Section, we describe Algorithm 1. A meta-algorithm that encompasses several existing imitation learning algorithms. Inside each iteration of the main for loop, the learner collects a new trajectory sampling actions from the policy $\pi^k$ (Line 4 in Algorithm 1) and then performs the following steps.

- **The Cost update.** At Line 6 of Algorithm 1, the learner updates an estimate of the true unknown cost function

---

[2]In order, to accommodate state-only and state-action with a unified analysis we overload the notation for the expert dataset. $\mathcal{D}_{\pi_E}$ denotes a collection of samples from the expert state occupancy measure in the former case and a collection of state-actions sampled from the state-action occupancy measure in the latter case.

---

**Algorithm 1** SOAR-Imitation Learning

---

**Require:** Reward step size $\alpha$, Expert dataset $\mathcal{D}_{\pi_E}$, Discount factor $\gamma$, Policy step size $\eta$.

1: Initialize $\pi^1$ as uniform distribution over $\mathcal{A}$.
2: Initialize empty replay buffer,i.e. $\mathcal{D}^0 = \{\}$
3: **for** $k = 1$ to $K$ **do**
4:    $\tau^k \leftarrow$ COLLECTTRAJECTORY$(\pi^k)$
5:    Add $\tau^k$ to replay buffer,i.e. $\mathcal{D}^k = \mathcal{D}^{k-1} \cup \tau^k$.
6:    $c^k \leftarrow$ UPDATECOST$(c^{k-1}, \mathcal{D}_{\pi_E}, \mathcal{D}^k, \alpha)$
7:    **for** $\ell = 1$ to $L$ **do**
8:      Compute estimator $Q_\ell^k$.
9:    **end for**
10:   $Q^k =$ OPTIMISTICQ$(\{Q_\ell^k\}_{\ell=1}^L)$.
11:   $\pi^k(a|s) =$ POLICYUPDATE$(\eta, \{Q^\tau(s,a)\}_{\tau=1}^k)$
12: **end for**

---

with the algorithm-dependent routine UPDATECOST. For instance, Generative Adversarial Imitation Learning (GAIL), Adversarial Inverse Reinforcement Learning (AIRL), and Discriminator Actor Critic (DAC) (Ho & Ermon, 2016; Fu et al., 2018; Kostrikov et al., 2019) use a reward derived from a discriminator neural network trained to distinguish state-action pairs visited by the expert from those visited by the learner.

Using a fixed cost function obtained from a behavioral cloning warm up is, instead, the approach taken in CSIL (Watson et al., 2023). Moreover, updating the reward to minimize an information theoretic divergence between expert and learner state occupancy measure is the approach taken in RKL (Ni et al., 2021). Finally, (Zeng et al., 2022) updates the cost using online gradient descent (OGD) (Zinkevich, 2003).

- **The state-action value function update.** In the *for* loop at Lines 7-9 of Algorithm 1, the learner updates $L$ different critics trained on different subsets of the data sampled from the replay buffer $\mathcal{D}^k$ denoted as $\{\mathcal{D}_\ell^k\}_{\ell=1}^L$. For a fixed state-action pair each dataset contains independent samples from $P(\cdot|s, a)$. This allows creating $L$ jointly independent random variables $\{Q_\ell^k\}_{\ell=1}^L$ that estimate the ideal value iteration update (i.e. $c^k + \gamma P V^k$) which cannot be implemented exactly due to the lack of knowledge on $P$.

  In Section 3.1, we provide an explicit way to compute $L$ slightly optimistic estimates for the tabular setting. Moreover, in the deep imitation learning experiments, we train $L$ different critics via temporal difference, as it is commonly done in Soft Actor Critic implementation ( see (Haarnoja et al., 2018) ).

  Finally, in Line 10 of Algorithm 1, the $L$ critics are aggregated to generate an estimate $Q^{k+1}$ which is, with high probability, optimistic i.e. $Q^{k+1} \leq c^k + \gamma P V^k$.

In other words, it underestimates the update that could have been performed by value iteration if the transition matrix $P$ was known to the learner. We provide aggregation routines that satisfy this requirement if $L$ is large enough.

- **The policy update** As the last step of each inner loop, the learner updates the policy using the optimistic state-action value function estimate. In the tabular case, we will instantiate the update using an online mirror descent (OMD) step (Beck & Teboulle, 2003; Nemirovskij & Yudin, 1983) (also known as the multiplicative weights update (Warmuth et al., 1997; Auer et al., 1995)). As it will be evident from Section 4, this update we can ensure that the KL divergence between consecutive policies is upper bounded in terms of the policy step size $\eta$. For the continuous state-action experiments, the online mirror descent is approximated via a gradient descent step on the SAC loss.

*Remark* 3.1. Notice that only one pair of critics is used in the implementation of SAC ($L = 1$) that serves as base RL algorithm for several commonly used IL algorithms ( GAIL (Ho & Ermon, 2016), AIRL (Fu et al., 2018), IQ-Learn (Garg et al., 2021), PPIL (Viano et al., 2022b), RKL (Ni et al., 2021) and ML-IRL (Zeng et al., 2022)). As proven in Corollary 4.11, $L = 1$ is not enough to ensure optimism, not even in the tabular case. In our experiments, we show that a value of $L$ larger than 1 is beneficial in all the MuJoCo environments we tested on.

### 3.1. Algorithm with guarantees in the tabular case

We consider an instance of Algorithm 1 in the tabular case for which we will prove theoretical sample efficiency guarantees. We present the pseudocode in Algorithm 2.

For what concerns the analysis, the first step is to extract the policy achieving the sample complexity guarantees above via an online-to-batch conversion. That is, the output policy is sampled uniformly from a collection of $K$ policies $\{\pi^k\}_{k=1}^K$. The sample complexity result follows from proving that the policies $\{\pi^k\}_{k=1}^K$ produced by Algorithm 2 is a sequence with sublinear regret in high probability. More formally, we define the regret as follows.

**Definition 3.2. Regret** The regret is defined as follows

$$\text{Regret}(K) \triangleq \frac{1}{1 - \gamma} \sum_{k=1}^K \left\langle c_{\text{true}}, d^{\pi^k} - d^{\pi_E} \right\rangle$$

*Remark* 3.3. Notice that the regret defined in this way satisfies $\text{Regret}(K) = \sum_{k=1}^K \left\langle \boldsymbol{\nu}_0, V_{c_{\text{true}}}^{\pi^k} - V_{c_{\text{true}}}^{\pi_E} \right\rangle$. For this reason, we require the factor $(1 - \gamma)^{-1}$ in the definition.

Omitting dependencies on the horizon and the state action

spaces cardinality, we will guarantee that

$$\text{Regret}(K) \leq \mathcal{O}(K^{1/2} + K |\mathcal{D}_{\pi_{\mathrm{E}}}|^{-1/2}),$$

with high probability. Notice that this bound is sublinear in $K$, for $|\mathcal{D}_{\pi_{\mathrm{E}}}| = \mathcal{O}(K)$. To obtain such bound, we adopt the following decomposition for $(1-\gamma)\text{Regret}(K)$ adapted from (Shani et al., 2021) to accommodate the infinite horizon setting.

$$\underbrace{\sum_{k=1}^{K} \left\langle c^k, d^{\pi^k} - d^{\pi_{\mathrm{E}}} \right\rangle}_{:= (1-\gamma)\text{Regret}_\pi(K, \pi_{\mathrm{E}})} + \underbrace{\sum_{k=1}^{K} \left\langle c_{\text{true}} - c^k, d^{\pi^k} - d^{\pi_{\mathrm{E}}} \right\rangle}_{:= (1-\gamma)\text{Regret}_c(K, c_{\text{true}})} \quad (1)$$

---

**Algorithm 2** Tabular SOAR-IL

---

**Require:** Step size $\eta$, Expert dataset $\mathcal{D}_{\pi_{\mathrm{E}}}$, Discount factor $\gamma$, Reward step size $\alpha$, $N^0(s,a) = 0$ for all $s,a$, number of estimators $L = 36 \log (|\mathcal{S}| |\mathcal{A}| K/\delta)$.
1: Initialize $\pi^1$ as uniform distribution over $\mathcal{A}$
2: **for** $k = 1$ to $K$ **do**
3:   Sample trajectory length $L^k \sim \text{Geometric}(1-\gamma)$.
4:   $\tau^k = \left\{ (s_t^k, a_t^k) \right\}_{t=1}^{L^k}$ rolling out $\pi^k$ for $L^k$ steps.
5:   Update counts for all $s_t^k, a_t^k \in \tau_k$:

$$N^k(s_t^k, a_t^k) = N^{k-1}(s_t^k, a_t^k) + 1.$$

6:   Add $s_t^k, a_t^k, s_{t+1}^k$ to the datasets with index $\ell = N^k(s_t^k, a_t^k) \mod L$,

$$\mathcal{D}_\ell^k = \mathcal{D}_\ell^{k-1} \cup \left\{ s_t^k, a_t^k \right\},$$
$$\mathcal{R}_\ell^k = \mathcal{R}_\ell^{k-1} \cup \left\{ s_t^k, a_t^k, s_{t+1}^k \right\}.$$

7:   $c^k = \text{COSTUPDATETABULAR}(c^{k-1}, \tau^k, \mathcal{D}_{\pi_{\mathrm{E}}})$.
8:   **for** $\ell = 1$ to $L$ **do**
9:     $N_\ell^k(s,a,s') = \sum_{\bar{s}, \bar{a}, \bar{s}' \in \mathcal{R}_\ell^k} \mathbb{1}_{\{\bar{s}, \bar{a}, \bar{s}' = s, a, s'\}}$.
10:     $N_\ell^k(s,a) = \sum_{\bar{s}, \bar{a} \in \mathcal{D}_\ell^k} \mathbb{1}_{\{\bar{s}, \bar{a} = s, a\}}$.
11:     $\widehat{P}_\ell^k(\cdot | s, a) = \frac{N_\ell^k(s,a,\cdot)}{N_\ell^k(s,a)+2}$
12:   **end for**
13:   $Q^{k+1} = \text{OPTQTAB}(V^k, \left\{ \widehat{P}_\ell^k \right\}_{\ell=1}^{L}, c^k)$.
14:   $\pi^{k+1}(a|s) \propto \pi^k(a|s) \exp\left( -\eta Q^{k+1}(s,a) \right)$
15:   $V^{k+1}(s) = \left\langle \pi^{k+1}(\cdot | s), Q^{k+1}(s, \cdot) \right\rangle$
16: **end for**
17: **Return** The mixture policy $\widehat{\pi}^K$.

---

The algorithmic design for the tabular setting aims at updating the cost variable so that the term $\text{Regret}_c$ grows sublinearly (see Line 7 in Algorithm 2).

We consider both cases of imitation from state-action expert data (Lines 4-6 of COSTUPDATETABULAR ) and state-only expert data (Lines 2-3 of COSTUPDATETABULAR).

These cases differ only in the stochastic loss for the cost update. Notice that we overload the notation to address both state-only and state-action imitation learning with a unified analysis. In particular, $\widehat{d^{\pi^k}}$ is an unbiased estimate of the learner occupancy measure. For state-only imitation learning we use $\widehat{d^{\pi^k}} = \mathbf{e}_{s_{L^k}^k}$ and estimated expert occupancy measure equals to $\widehat{d^{\pi_{\mathrm{E}}}} = |\mathcal{D}_{\pi_{\mathrm{E}}}|^{-1} \sum_{s \in \mathcal{D}_{\pi_{\mathrm{E}}}} \mathbf{e}_s$ while for state-action imitation learning $\widehat{d^{\pi^k}} = \mathbf{e}_{s_{L^k}^k, a_{L^k}^k}$ and $\widehat{d^{\pi_{\mathrm{E}}}} = |\mathcal{D}_{\pi_{\mathrm{E}}}|^{-1} \sum_{s,a \in \mathcal{D}_{\pi_{\mathrm{E}}}} \mathbf{e}_{s,a}$. The formal bound on $\text{Regret}_c$ is given in Theorem 4.3. The rest of the algorithm

---

**Algorithm 3** COSTUPDATETABULAR

---

**Require:** Current cost vector $c^{k-1}$, trajectory $\tau^k$, expert dataset $\mathcal{D}_{\pi_{\mathrm{E}}}$.
1: **if** STATE-ONLY = TRUE **then**
2:   $\widehat{d^{\pi^k}} = \mathbf{e}_{s_{L^k}^k}$.
3:   $\widehat{d^{\pi_{\mathrm{E}}}} = |\mathcal{D}_{\pi_{\mathrm{E}}}|^{-1} \sum_{s \in \mathcal{D}_{\pi_{\mathrm{E}}}} \mathbf{e}_s$
4: **else**
5:   $\widehat{d^{\pi^k}} = \mathbf{e}_{s_{L^k}^k, a_{L^k}^k}$.
6:   $\widehat{d^{\pi_{\mathrm{E}}}} = |\mathcal{D}_{\pi_{\mathrm{E}}}|^{-1} \sum_{s,a \in \mathcal{D}_{\pi_{\mathrm{E}}}} \mathbf{e}_{s,a}$
7: **end if**
8: **Return:** $c^k \leftarrow \Pi_{\mathcal{C}} \left[ c^{k-1} - \alpha(\widehat{d^{\pi_{\mathrm{E}}}} - \widehat{d^{\pi^k}}) \right]$

---

aims to provide a sublinear bound on $\text{Regret}_\pi$. In particular, the updates for the estimated transition kernels $\left\{ \widehat{P}_\ell^k \right\}_{\ell=1}^{L}$ in Lines 8-12 of Algorithm 2 serves to build $L$ slightly optimistic [3] estimate of the ideal value function update.

In the routine OPTIMISTICQTABULAR, we propose two aggregation rules to generate the optimistic $Q$ value estimate to be used in the policy update step. The first one, takes the minimum of the $L$ estimators as in Equation (Min), while the second option Equation (Mean-Std) considers the mean of the $L$ estimators minus a factor proportional to the empirical standard deviation. By Samuelson's inequality (Samuelson, 1968), we prove that the second option is more optimistic.

Finally, an iteration of the tabular case algorithm is concluded by the policy update implemented via OMD.

Having described our main techniques we are in the position of stating our main theoretical results hereafter.

**Theorem 3.4.** *Main Result For any MDP, let us consider either the update Equation* (Min) *or Equation* (Mean-Std)*, it holds that with probability $1 - 5\delta$ that $\frac{\text{Regret}(K)}{K}$ of Tabular*

---

[3]The optimism is achieved by adding 2 in the denominator of the estimated transition kernels.

**Algorithm 4** OPTIMISTICQTABULAR ( OPTQTAB)

---

**Require:** current state value function estimate $V^k$, ensemble of estimated transitions $\left\{ \widehat{P}_\ell^k \right\}_{\ell=1}^L$, cost $c^k$.

1: // Option 1

$$\textbf{return} \quad Q^{k+1} = c^k + \gamma \min_{\ell \in [L]} \widehat{P}_\ell^k V^k \qquad \text{(Min)}$$

2: // Option 2

$$\textbf{return} \quad Q^{k+1} = c^k + \gamma \max \left[ \frac{1}{L} \sum_{\ell=1}^L \widehat{P}_\ell^k V^k - \sigma^k, 0 \right]$$
(Mean-Std)

3: with $\sigma^k = \sqrt{\sum_{\ell=1}^L \left( \widehat{P}_\ell^k V^k - \frac{1}{L} \sum_{\ell'=1}^L \widehat{P}_{\ell'}^k V^k \right)^2}$.

---

*SOAR-IL (Algorithm 2) is upper bounded by*

$$\widetilde{\mathcal{O}} \left( \sqrt{\frac{|\mathcal{S}|^4 |\mathcal{A}| \log(1/\delta)}{(1-\gamma)^5 K}} \right)$$
$$+ \sqrt{\frac{|\mathcal{S}|^2 |\mathcal{A}| \log \left( |\mathcal{S}| |\mathcal{A}| /\delta \right) (\log(|\mathcal{S}|)+2)^2}{(1-\gamma)^2 |\mathcal{D}_{\pi_E}|}}.$$

*Therefore, choosing $K = \widetilde{\mathcal{O}} \left( \frac{|\mathcal{S}|^4 |\mathcal{A}| \log(1/\delta)}{(1-\gamma)^5 \epsilon^2} \right)$ and $|\mathcal{D}_{\pi_E}| = \frac{|\mathcal{S}|^2 |\mathcal{A}| \log(|\mathcal{S}||\mathcal{A}|/\delta)(\log(|\mathcal{S}|)+2)^2}{\epsilon^2(1-\gamma)^2}$ it holds that the mixture policy $\widehat{\pi}_K$ satisfies $\left\langle \boldsymbol{\nu}_0, V_{c_{\text{true}}}^{\widehat{\pi}^k} - V_{c_{\text{true}}}^{\pi_E} \right\rangle \leq \epsilon$ with probability at least $1 - 5\delta$.*

*Remark* 3.5. The bound on $|\mathcal{D}_{\pi_E}|$ is the bound on the number of either state-only or state-action expert trajectories depending on the setting considered.

*Remark* 3.6. The gurantees are stated for the mixture policy $\widehat{\pi}^K$, i.e. the policy which has an occupancy measure equal to the average occupancy measure of the policies in the no-regret sequence. That is, it holds that $d^{\widehat{\pi}^K} = K^{-1} \sum_{k=1}^K d^{\pi^k}$. The policy $\widehat{\pi}^K$ cannot be computed without knowledge of $P$ but sampling a trajectory from it can be done by choosing an index $k \sim \text{Unif}([K])$ at the beginning of each new episode and continuing rolling out the policy $\pi^k$ for a number of steps sampled from $\text{Geom}(1-\gamma)$.

*Remark* 3.7. In the case of state-only expert dataset the provided upper bounds for $K$ and $|\mathcal{D}_{\pi_E}|$ are optimal up to log factors in the precision parameters $\epsilon$. Indeed, these upper bounds match the lower bounds in (Moulin et al., 2025).

## 4. Theoretical analysis

We need to start with an important remark on the structure of the MDP considered in the proof.

*Remark* 4.1. For technical reasons, in particular for the

proof of Corollary 4.11, we consider as intermediate step in the proof MDPs where from each state action pairs is possible to observe a transition to only two other possible states. While this restriction on the dynamics appears to be limiting any MDP can be cast into this form at the cost of a quadratic blow up in the number of states, from $|\mathcal{S}|$ to $|\mathcal{S}|^2$. To see this, for a general MDP where from a given state action pair a transition to all possible $|\mathcal{S}|$ states can be observed is equivalent to a binarized MDP where this *one layer* transition is represented with a tree of depth at most $\log_2(|\mathcal{S}|)$ with binary transitions only. Moreover, the discount factor in the binarized MDP should be set to $\gamma_{\text{bin}} = \gamma^{-\log_2 |\mathcal{S}|}$ to maintain the return unchanged. We consider in this section a binarized MDP with $|\mathcal{S}|$ states in this section and we squared the number of states in stating Theorem 3.4 which holds for general MDPs. Moreover, in stating the result for general MDP we also inflated the effective horizon by a factor $\log_2 |\mathcal{S}|$ as shown in Lemma C.6.

As mentioned, the proof is decomposed into two main parts: (i) bounding the policy regret $\text{Regret}_\pi$ and (ii) bounding the cost updates regret $\text{Regret}_c$. In particular, we can prove the two following results.

**Theorem 4.2.** *Policy Regret In a binarized MDP with $|\mathcal{S}|$ states and discount factor $\gamma$, it holds that with probability $1 - 3\delta$, for any policy $\pi^\star$, $\text{Regret}_\pi(K, \pi^\star)$ is upper bounded by*

$$\frac{\log |\mathcal{A}|}{\eta(1-\gamma)} + \frac{\eta K}{(1-\gamma)^4} + \widetilde{\mathcal{O}} \left( \frac{\sqrt{K |\mathcal{S}|^2 |\mathcal{A}| \log(1/\delta)}}{(1-\gamma)^2} \right)$$

*and for $\eta = \sqrt{\frac{\log |\mathcal{A}| (1-\gamma)^3}{K}}$ it holds that using the update in (Min) or in (Mean-Std) it holds that $\text{Regret}(K, \pi^\star)$ is upper bounded by $\widetilde{\mathcal{O}} \left( \sqrt{\frac{K |\mathcal{S}|^2 |\mathcal{A}| \log(1/\delta)}{(1-\gamma)^5}} \right)$.*

**Theorem 4.3.** *Cost Regret In a binarized MDP with $|\mathcal{S}|$ states and discount factor $\gamma$, it holds that with probability $1 - 2\delta$, $(1-\gamma)\text{Regret}_c(K; c_{\text{true}})$ is upper bounded by*

$$4\sqrt{K \log(1/\delta)} + K \sqrt{\frac{|\mathcal{S}| |\mathcal{A}| \log \left( |\mathcal{S}| |\mathcal{A}| /\delta \right)}{2 |\mathcal{D}_{\pi_E}|}}$$

*Remark* 4.4. Once Theorems 4.2 and 4.3 are proven the bound on Theorem 3.4 follows trivially by a union bound and bounding $\text{Regret}_\pi$ and $\text{Regret}_c$ with Theorem 4.2 and Theorem 4.3 respectively and dividing everything by $K$ (because in Theorem 3.4 we consider the quantity $\text{Regret}(K)/K$). Finally, we also divide by $1 - \gamma$, to match the definition of $\text{Regret}(K)$ in Definition 3.2.

### 4.1. Proof Sketch of Theorem 4.2

The regret decomposition towards the proof of Theorem 4.2 leverages the following Lemma.

**Lemma 4.5.** *Consider the MDP* $M = (\mathcal{S}, \mathcal{A}, P, c, \boldsymbol{\nu}_0, \gamma)$ *and two policies* $\pi, \pi' : \mathcal{S} \to \Delta_{\mathcal{A}}$. *Then consider for any* $\widehat{Q} \in \mathbb{R}^{|\mathcal{S}||\mathcal{A}|}$ *and* $\widehat{V}^{\pi}(s) = \left\langle \pi(\cdot|s), \widehat{Q}(s, \cdot) \right\rangle$ *and* $Q^{\pi'}, V^{\pi'}$ *be respectively the state-action and state value function of the policy* $\pi$ *in MDP* $M$. *Then, it holds that* $(1 - \gamma) \left\langle \boldsymbol{\nu}_0, \widehat{V}^{\pi} - V^{\pi'} \right\rangle$ *equals*

$$\left\langle d^{\pi'}, \widehat{Q} - c - \gamma P \widehat{V}^{\pi} \right\rangle + \mathbb{E}_{s \sim d^{\pi'}} \left[ \left\langle \widehat{Q}(s, \cdot, \pi(\cdot|s) - \pi'(\cdot|s) \right\rangle \right].$$

*Remark* 4.6. This Lemma is a generalization of the well-known performance difference Lemma (Kakade, 2001) to the case of inexact value functions. Indeed, notice that if $\widehat{Q} = Q^{\pi}$, then the first term in the decomposition equals zero and the result boils down to the standard performance difference Lemma. For arbitrary $\widehat{Q}$, the first term is a temporal difference error averaged by the occupancy measure $d^{\pi'}$.

We can apply two times Lemma 4.5 on each of the summands of the sum from $k = 1$ to $K$, to obtain a convenient decomposition of $\mathrm{Regret}_{\pi}$. Denoting $\delta^k(s, a) \triangleq c^k(s, a) + \gamma P V^k(s, a) - Q^{k+1}(s, a)$ and $g^k(s, a) \triangleq Q^{k+1}(s, a) - Q^k(s, a)$, we have that

$$(1 - \gamma) \mathrm{Regret}_{\pi}(K; \pi^{\star}) =$$

$$(1 - \gamma) \sum_{k=1}^{K} \left\langle \boldsymbol{\nu}_0, V_{c^k}^{\pi^k} - V^k + V^k - V_{c^k}^{\pi^{\star}} \right\rangle =$$

$$\sum_{k=1}^{K} \mathbb{E}_{s \sim d^{\pi^{\star}}} \left[ \left\langle Q^k(s, \cdot), \pi^k(s) - \pi^{\star}(s) \right\rangle \right] \qquad \text{(BTRL)}$$

$$+ \sum_{k=1}^{K} \sum_{s,a} \left[ d^{\pi^k}(s, a) - d^{\pi^{\star}}(s, a) \right] \cdot \left[ \delta^k(s, a) \right]$$

$$\text{(Optimism)}$$

$$+ \sum_{k=1}^{K} \mathbb{E}_{s,a \sim d^{\pi^k}} \left[ g^k(s, a) \right] - \sum_{k=1}^{K} \mathbb{E}_{s,a \sim d^{\pi^{\star}}} \left[ g^k(s, a) \right]$$

$$\text{(Shift)}$$

Next, we bound each of these terms individually. Starting from the first term, the next Lemma shows that our policy update (Line 14 of Algorithm 2) can be seen as an instance of Be the regularized leader (BTRL) ( see e.g. (Orabona, 2023) ). Therefore, it guarantees that for any sequence $\{Q^k\}_{k=1}^{K}$, the term (BTRL) is bounded as follows.

**Lemma 4.7.** *Let us consider the sequence of policies* $\{\pi^k\}_{k=1}^{K}$ *generated by Algorithm 2 for all* $\eta > 0$ *then it holds that* $\text{(BTRL)} \leq \frac{\log|\mathcal{A}|}{\eta}$.

Next, we show that thanks to the multiplicative weights update for the policy the KL divergence between consecutive policies is upper bounded by the policy step size $\eta$, i.e.

$D_{KL}(\pi^{k+1}(\cdot|s), \pi^k(\cdot|s)) \leq \mathcal{O}(\eta)$ for all $s \in \mathcal{S}$. Thanks to this *slow changing* property, we can prove the following bound on (Shift).

**Lemma 4.8.** *For the sequence of policies* $\{\pi^k\}_{k=1}^{K}$ *generated by Algorithm 2, for all* $\eta > 0$, *it holds that* (Shift) $\leq \frac{\eta K}{(1-\gamma)^3}$.

*Remark* 4.9. The step size choice for $\eta$ in Theorem 4.2 is made to trade off optimally the bounds in Lemmas 4.7 and 4.8.

Finally, the most technical part of the proof aims at bounding the term (Optimism).

**Lemma 4.10.** *Let us consider an MDP where* $\max_{s,a \in \mathcal{S} \times \mathcal{A}} \mathrm{supp}(P(\cdot|s, a)) = 2$. *For each* $k \in [K]$, *if the* $Q^{k+1}$ *in Algorithm 2, are updated according to* (Min) *or* (Mean-Std), *the iterates produced by Algorithm 2 satisfy with probability* $1 - 3\delta$ *that*

$$\text{(Optimism)} \leq \widetilde{\mathcal{O}} \left( \frac{\sqrt{K |\mathcal{S}|^2 |\mathcal{A}| \log(1/\delta)}}{1 - \gamma} \right).$$

**Proof sketch of Lemma 4.10** The proof of this Lemma, leverages that the temporal difference errors $\delta^k(s, a)$ produced by Algorithm 2 are positive with high probability as shown by the next result[4].

**Corollary 4.11.** *Consider an MDP where* $\max_{s,a \in \mathcal{S} \times \mathcal{A}} \mathrm{supp}(P(\cdot|s, a)) = 2$, *then for* $L \geq 36 \log \left( \frac{|\mathcal{S}||\mathcal{A}|K}{\delta} \right)$ *it holds that with probability at least* $1 - \delta$

$$\min_{\ell \in [L]} \widehat{P}_{\ell}^k V^k(s, a) \leq P V^k(s, a) \ \forall \ s, a \in \mathcal{S} \times \mathcal{A}, \ \forall \ k \in [K].$$

Corollary 4.11 implies that $-\left\langle d^{\pi^{\star}}, \delta^k \right\rangle \leq 0$ for all $k \in [K]$ and therefore that (Optimism) $\leq \sum_{k=1}^{K} \left\langle d^{\pi^k}, \delta^k \right\rangle$.

*Remark* 4.12. The above inequality, it is crucial for obtaining the result. Indeed, it upper bounds (Optimism) with the *on-policy* temporal difference errors [5] which are small enough to ensure sublinear regret. To see this (informally) consider two cases. First, let us assume that $d^{\pi^k}$ is relatively large for some action pair. Then, that action pair is expected to be visited often in the rollouts and therefore $\delta^k$ is expected to be small. Vice versa, if $\delta^k$ for a certain state-action pair is large, this means that for that state-action pair $d^{\pi^k}$ is relatively small. Overall, we always expect the product $\left\langle d^{\pi^k}, \delta^k \right\rangle$ to be a small quantity. Notice that the

---

[4]In the main text, we present the proof for the update in (Min). The case of update as in (Mean-Std) is deferred to the Appendix.

[5]That is the temporal difference errors $\delta^k$ averaged by the learner occupancy measures $d^{\pi^k}$

same arguments could not have been carried out replacing $d^{\pi^k}$ with $d^{\pi^\star}$ because the rollouts used in Algorithm 2 are not sampled with $\pi^\star$.

To formalize the above intuition, we upper bound the temporal difference errors with the inverse of the number of times each state-action pair is visited.

**Lemma 4.13.** *(Simplified Version of Lemma B.5 ) Let us consider a binarized MDP with $|\mathcal{S}|$ states and discount factor $\gamma$. With probability $1 - \delta$, it holds that for all $s, a \in \mathcal{S} \times \mathcal{A}$ and for all $k \in [K]$,*

$$\delta^k(s,a) \leq \widetilde{\mathcal{O}}\left(\sqrt{\frac{L\,|\mathcal{S}|\log(1/\delta)}{(N^k(s,a)+1)(1-\gamma)^2}}\right).$$

Therefore, by concentration inequalities and noticing that $s_{L^k}^k, a_{L^k}^k \sim d^{\pi^k}$, it holds that with high probability

$$\sum_{k=1}^K \left\langle d^{\pi^k}, \delta^k \right\rangle = \widetilde{\mathcal{O}}\left(\sum_{k=1}^K \delta^k(s_{L^k}^k, a_{L^k}^k)\right)$$
$$\leq \widetilde{\mathcal{O}}\left(\sum_{k=1}^K \sqrt{\frac{L\,|\mathcal{S}|\log(1/\delta)}{(N^k(s_{L^k}^k, a_{L^k}^k)+1)(1-\gamma)^2}}\right)$$
$$\leq \widetilde{\mathcal{O}}\left(\sqrt{K\sum_{k=1}^K \frac{L\,|\mathcal{S}|\log(1/\delta)}{(N^k(s_{L^k}^k, a_{L^k}^k)+1)(1-\gamma)^2}}\right).$$

At this point, the proof is concluded by bounding the last sum over $K$ with a standard numerical sequences argument (see Lemma C.2).

**Optimal choice of the number of critics network $L$** It is important to notice that Corollary 4.11 and Lemma 4.13 creates a tradeoff for what concerns the optimal choice of the number of critics. In particular, from Corollary 4.11, $L$ should be chosen large enough to ensure that optimism holds with high enough probability. On the other hand, one can notice that Lemma 4.13 upper bounds the expected on policy temporal difference error as $\mathcal{O}(L)$ therefore a smaller number of critics ensures a tighter bound. All in all, the best choice is the smallest $L$ that ensures optimism with probability at least $1 - \delta$, that is $L = 36\log\left(\frac{|\mathcal{S}||\mathcal{A}|K}{\delta}\right)$. The tradeoff with respect to the number of critics is also observed in a practical ablation study (see Figures 5 and 4) .

### 4.2. Proof Sketch of Theorem 4.3

The proof of this term is considerably easier than the bound of the regret for the policy player because we have exact knowledge of the decision variables domain [6]. The first step

---

[6] $\mathcal{C}$ is taken to be the $\ell_\infty$-ball of radius 1

in the proof is to decompose $(1 - \gamma)\mathrm{Regret}_c$ as follows

$$\sum_{k=1}^K \left\langle c_{\mathrm{true}} - c^k, \widehat{d^{\pi^k}} - \widehat{d^{\pi_E}} \right\rangle$$
$$+ \sum_{k=1}^K \left\langle c_{\mathrm{true}} - c^k, d^{\pi^k} - \widehat{d^{\pi^k}} \right\rangle$$
$$+ \sum_{k=1}^K \left\langle c_{\mathrm{true}} - c^k, \widehat{d^{\pi_E}} - d^{\pi_E} \right\rangle.$$

The first term in the decomposition is upper bounded by $\mathcal{O}(\sqrt{K})$ via a standard online gradient descent analysis (Zinkevich, 2003). Since $\widehat{d^{\pi^k}}$ is an unbiased estimate of the learner occupancy measure, the second term in the decomposition is the sum of a martingale difference sequence. Therefore, an application of the Azuma-Hoeffding inequality ensures that this term grows as $\widetilde{\mathcal{O}}\left(\log(1/\delta)\sqrt{K}\right)$ with probability at least $1 - \delta$.

Finally, the last term is bounded as $\widetilde{\mathcal{O}}\left(K\log(1/\delta)\,|\mathcal{D}_{\pi_E}|^{-1/2}\right)$ with probability at least $1 - \delta$. This is done, proving that for the empirical average estimators for the expert occupancy measure it holds that $\left\|d^{\pi_E} - \widehat{d^{\pi_E}}\right\|_1 \leq |\mathcal{D}_{\pi_E}|^{-1/2}\log(1/\delta)$ with probability at least $1 - \delta$. A union bound concludes the proof of Theorem 4.3. The formal proof is deferred to the Appendix.

## 5. SOAR for continuous state and actions problems.

In this section, we explain how Algorithm 1 is instantiated in imitation learning problems with continuous states and action spaces, which therefore requires neural networks to approximate the value function and policy updates. Since in our analysis for the tabular case, we need to use multiplicative weights/softmax updates, we decided to use SAC, which is an approximation of such updates in the continuous state-action setting.

However, the standard SAC keeps only one network, often called the critic network, to estimate the $Q$ values. On the other hand, we use a pair of them to avoid the excessive overestimation noticed in Double DQN (van Hasselt et al., 2015). Since it uses only one pair of critics, SAC cannot achieve optimism reliably with high probability.

To fix this issue, we consider multiple critics and we used as an optimistic estimate the mean minus the standard deviation of the ensemble as explained in Algorithm 5. In addition, the standard deviation needs to be truncated at a threshold, as was done in the tabular analysis, to avoid the value function estimators growing out of the attainable range. For any state $s$, the estimated value functions are

truncated in the interval $\left[0, (1-\gamma)^{-1}\right]$. Each of the esti-

---

**Algorithm 5** OPTIMISTICQ-NN

---
**Require:** Replay buffer $\mathcal{D}$, Estimators $\{Q_\ell\}_{\ell=1}^L$, maximum
    standard deviation $\sigma$.
1: $\{s_i\}_{i=1}^N \leftarrow$ sample observations from $\mathcal{D}$
2: $a_i \leftarrow \pi(s_i)$
3: $\bar{Q}(s_i, a_i) = \frac{1}{L}\sum_{\ell=1}^L Q_\ell(s_i, a_i)$
4: std-Q$(s_\ell, a_\ell) = \sqrt{\frac{1}{L}\sum_{\ell=1}^L \left(Q_\ell(s_i, a_i) - \bar{Q}(s_i, a_i)\right)^2}$
5: $\overline{\text{std-Q}}(s_i, a_i) \leftarrow \text{Clip}(\text{std-Q}(s_i, a_i), 0, \sigma)$.
6: $Q(s_i, a_i) = \bar{Q}(s_i, a_i) - \overline{\text{std-Q}}(s_i, a_i)$
7: **Return:** $Q(s_i, a_i)$ for all $i = 1, \ldots, N$.

---

mators (critics) $\{Q_\ell\}_{\ell=1}^L$ is trained in the same way (minimizing the squared Bellman error as in standard SAC ) on a different dataset collected by the same actor. That is, on independent identically distributed datasets. For completeness, the SAC critic training is included in Algorithm 7 in Appendix F. In the continuous setting, it is clearly not possible to compute the optimistic state-action value at every state-action pair. Thankfully, it suffices to compute the optimistic state action value function $Q$, invoking the routine OPTIMISTICQ-NN, only for the state-actions in a mini-batch $\mathcal{D} = \{s_i, a_i\}_{i=1}^N$. Indeed, the policy network weights does not require perfect knowledge of $Q$ over $\mathcal{S} \times \mathcal{A}$ but only an Adam (Kingma & Ba, 2015) update step on the loss $\mathcal{L}_\pi = \frac{1}{N}\sum_{i=1}^N \left(-\eta \log \pi(a_i|s_i) + Q(s_i, a_i)\right)$.

In the next section, we show that for multiple choices of UPDATECOST (ML-IRL, CSIL and RKL) replacing the standard SAC critic update routine with OPTIMISTICQ-NN leads to improved performance.

# 6. Experiments

We perform experiments for both state only and state action IL on the following MuJoCo (Todorov et al., 2012) environments: Ant, Hopper, Walker2d, and Humanoid.

For the state-only IL setting, we showcase the improvement on RKL (Ni et al., 2021) and ML-IRL (State-Only) (Zeng et al., 2022). In both cases, we found that using $L = 4$ critic networks and an appropriately chosen value for the standard deviation clipping threshold $\sigma$ consistently improves upon the baseline. In the Appendix D, we conduct an ablation study for $L$ and $\sigma$.

We denote our derived algorithms as RKL+SOAR and ML-IRL+SOAR. In addition to observing an improvement over standard RKL and ML-IRL, we outperform the state-only version of the recently introduced OPT-AIL algorithm (Xu et al., 2024) (see Figure 2) which incorporates an alternative, more complicated, deep exploration technique.

For the state-action experiments, we plug in the SOAR template on CSIL, the state-action version of ML-IRL and HYPE (Ren et al., 2024). We coined the derived versions CSIL+SOAR and ML-IRL+SOAR (see Appendix F for detailed pseudocodes of these algorithms). We also compare with GAIL (Ho & Ermon, 2016), SQIL (Reddy et al., 2019b), and OPT-AIL. We observe that the exploration mechanism injected by the SOAR principle allows us to achieve reliably superior results (see Figure 3).

Further details about the hyperparameters are provided in the Appendix D. Moreover, we notice that for all the algorithms in the higher-dimensional and thus more challenging environments (Ant-v5 and Humanoid-v5), the advantage of the SOAR exploration technique becomes more evident.

## 6.1. Experiment on a hard exploration task

An anonymous reviewer pointed out that the MuJoCo benchmark is not the hardest for what concern exploration. This is a very valid suggestion that we address here. This will also allow to understand better the role of the number of critics $L$. Therefore, to highlight even more the importance of exploration especially in imitation learning from states only we run SOAR-IL in the worst case construction used in the lower bound for the number of environment interaction in (Moulin et al., 2025, Theorem 19).

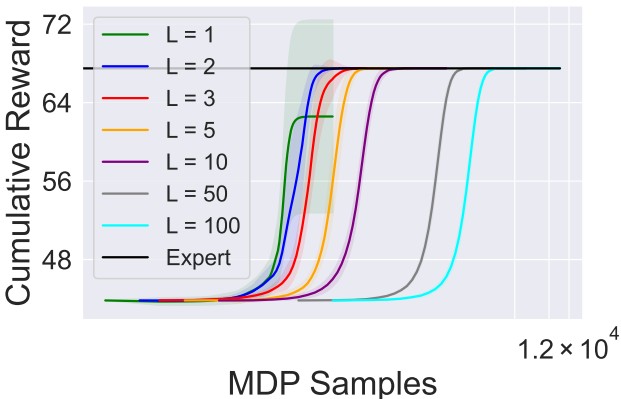

Figure 4: **Ablation for $L$ on hard exploration task**. State only imitation experiment in a hard exploration environment (used in the lower bound from (Moulin et al., 2025, Theorem 19)) . Results averaged over 5 seeds, for a dataset of 100 states sampled from the expert occupancy measure.

We can see that with only 1 network, the mean of the learner does not reach the expert performance and the variance is very high meaning that some seeds are successful and others fail. This is in perfect agreement with Corollary 4.11 which predicts that for low values of the number of critics $L$, the optimistic properties of the critic estimators can not be guaranteed with high probability. For $L = 2$, the environment

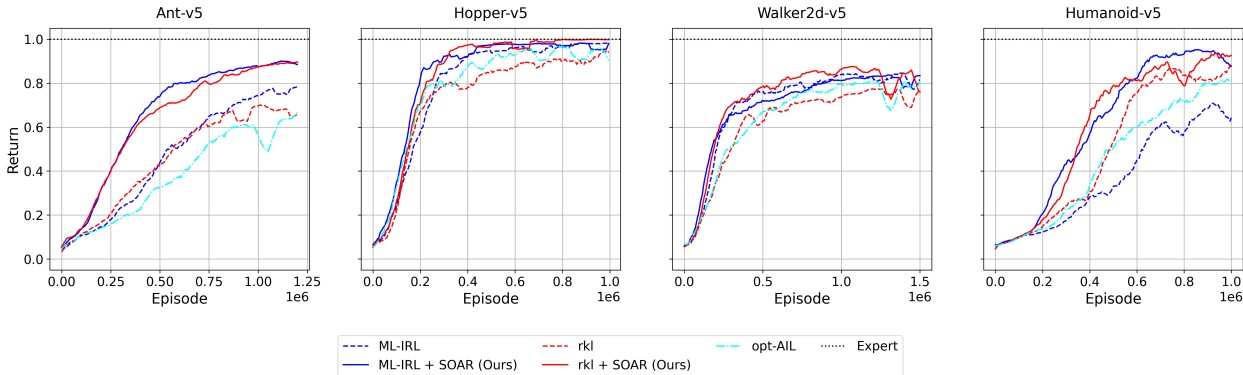

Figure 2: **Experiments from State-Only Expert Trajectories**. 16 expert trajectories, average over 5 seeds, $L = 4$ Clipping values $\sigma$ - ML-IRL: [Ant: 10.0, Hopper: 50.0, Walker2d: 0.5, Humanoid: 5.0], rkl: [Ant: 0.8, Hopper: 50.0, Walker2d: 30.0, Humanoid: 100.0]

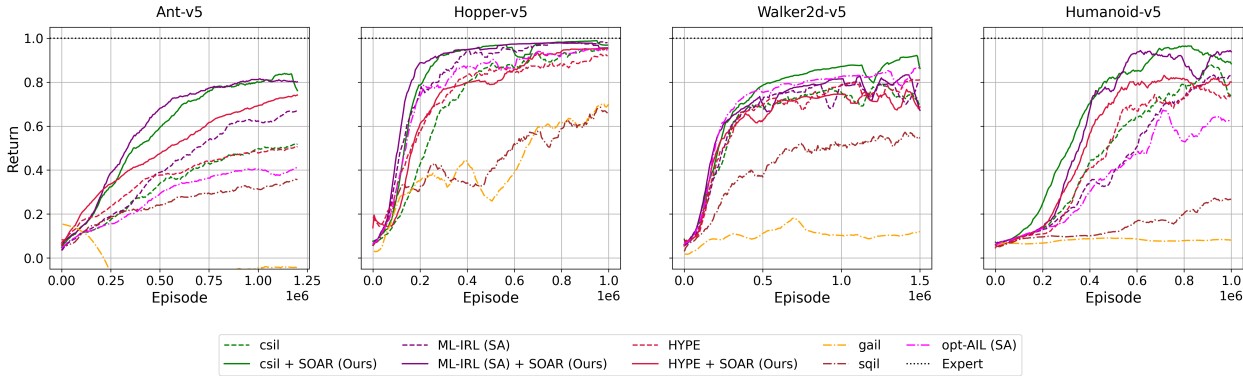

Figure 3: **Experiments from State-Action Expert Trajectories**. 16 expert trajectories, average over 5 seeds, $L = 4$. Clipping values $\sigma$ - CSIL: [Ant: 10.0, Hopper: 5.0, Walker2d: 0.5, Humanoid: 0.1], ML-IRL(SA): [Ant: 5.0, Hopper: 10.0, Walker2d: 0.5, Humanoid: 50.0]

is solved successfully albeit with a higher variance than the case $L = 3$.

Increasing $L$ further leads to worst results in terms of MDP samples needed to solve the task. This is because according to Lemma 4.13 the upper bound on the expected on policy temporal difference error scales with $L$ so an excessively large $L$ should be avoided. We used $\alpha = 0.5$, $\eta = 4$, and we scaled the standard deviation bonus by 0.001.

## 7. Conclusions and Open Questions

While there has been interest in developing heuristically effective exploration techniques in deep RL, the same is not true for deep IL. For example, even in the detailed study *What matters in Adversarial Imitation Learning ?* (Orsini et al., 2021) the effectiveness of deep exploration techniques is not investigated.

Prior to our work, only few studied the benefits of exploration in imitation learning, mostly in the state-only regime (Kidambi et al., 2021). However, their theoretical algorithm uses bonuses that cannot be implemented with neural

networks. Similarly, the recent work (Xu et al., 2024) uses exploration technique in Deep IL but requires solving a complicated non-concave maximization problem. Our approach is remarkably easier to implement. It achieves convincing empirical results results and enjoys theoretical guarantees.

Moreover, our framework can be applied to and is expected to be beneficial for other existing IL algorithms based on SAC (such as AdRIL (Swamy et al., 2021) and SMILING (Wu et al., 2024)). Moreover, the same hope for any future deep IL algorithm using SAC for policy updates.

**Open Questions** On the theoretical side, we plan to analyze the ensemble exploration technique with function approximation under more general structural assumption on the environment. From the practical one, a very relevant question is to investigate if the exploration enhanced versions of DQN (Osband et al., 2016a; 2018) can speed up imitation learning from visual input. Finally, the same idea might find application in the LLM finetuning given the recently highlighted potential of IL for this task (Wulfmeier et al., 2024; Foster et al., 2024).

## Acknowledgments

This work is funded (in part) through a PhD fellowship of the Swiss Data Science Center, a joint venture between EPFL and ETH Zurich. This work was supported by Hasler Foundation Program: Hasler Responsible AI (project number 21043). Research was sponsored by the Army Research Office and was accomplished under Grant Number W911NF-24-1-0048. This work was supported by the Swiss National Science Foundation (SNSF) under grant number 200021_205011.

## Impact Statement

The current submission is expected to have an impact in the imitation learning community particularly because it highlights the benefits of exploration, not only theoretically but also in simulated robotics experiments. Impact can be expected also in the broader machine learning community and in related disciplines such as robotics and control theory. Beyond that, we do not expect direct impact on the society (i.e. outside the machine learning community and IT industry ).

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

# A. Related Works

**IL Theory** The first theoretical guarantees obtained for the imitation learning problem dates back to the work of (Abbeel & Ng, 2004) and (Syed & Schapire, 2007) which notably used the idea of no-regret learning. However, their work requires either knowledge of the environment transitions or they require a suboptimal in the precision parameter $\varepsilon$ amount of expert trajectories to estimate those. Our theoretical guarantees are in the same setting of previous works like (Shani et al., 2021) and (Xu et al., 2023) which do not require knowledge of the environment transitions a priori but assumes online trajectory access to the environment. The main difference that their work focuses on the easier finite horizon setting. Additionally, their exploration techniques only applies to the tabular setting. Indeed, in the MuJoCo experiments in (Shani et al., 2021), the authors do not attempt to implement the exploration mechanism required for their theoretical guarantees. In a similar way, also (Xu et al., 2023) can not be implemented beyond the tabular setting because it relies on a reward free procedure requiring bonuses proportional to the number of visits to each state action pair. (Ren et al., 2024) suggest an algorithm which does not require exploration but it can not improve upon behavioural cloning in terms of expert trajectories. (Rajaraman et al., 2020; 2021b; Foster et al., 2024) analyze instead offline imitation learning (behavioural cloning) where no additional interaction with the environment is allowed. This setting is more general but it comes at the cost of additional assumptions such as policy realizability or worst depedence on the horizon on the required number of trajectories. (Foster et al., 2024) presents an analysis for general policy classes but they require a maximum likelihood oracle which can not be implemented exactly when using neural network function approximation.

There has been also a variety of studies tackling the problem of computationally efficient algorithm with linear function approximation such as (Kamoutsi et al., 2021; Viano et al., 2022b; 2024; Rajaraman et al., 2021a; Swamy et al., 2022). However, their proof techniques are strictly depending on the linearity of the dynamics therefore the experiments in continuous control tasks require changes in the algorithmic design. Albeit our guarantees are restricted to the tabular setting, the algorithm can be implemented with no modifications with neural networks.

Several works focus on the setting where expert queries are allowed at any state visited during the MDP interaction (Ross & Bagnell, 2010; Ross et al., 2011; Swamy et al., 2021) or that require a generative model for the algorithm updates (Swamy et al., 2022). Another recent work requires a generative model to sample the initial state of the trajectory from the expert occupancy measure (Swamy et al., 2023). Our algorithm requires sampling only trajectories in the MDP therefore it does not leverage the aforementioned generative model assumption. In contrast, the setting of this work matches the most practical one adopted for example in (Ho et al., 2016; Ho & Ermon, 2016; Fu et al., 2018; Reddy et al., 2019a; Dadashi et al., 2021; Watson et al., 2023; Garg et al., 2021; Ni et al., 2021). In this case, the expert policy can not be queried and the learner access only a precollected dataset of expert demonstrations.

**Theory for IL from States Only** This setting has been firstly studied in (Sun et al., 2019b) in the finite horizon setting and with general function approximation their work does not use exploration mechanism. However their work requires an additional realizability assumption of the expert value function, it can only learn a difficult to store and deploy non stationary policy and provides suboptimal guarantees on $\tau_E$ in terms of the horizon dependence.

The follow up from (Arora et al., 2020), still requires the realizability of the state value function which is not needed in our work. The work of Kidambi et al. (2021) uses the idea of exploration in state only finite horizon imitation learning. Their analysis for tabular MDP gives a bound on $K$ which has a worst horizon dependence and it requires the design of exploration bonuses tight to the structural properties of the MDP. Therefore, their NN experiments requires an empirical approximation of such bonuses while the SOAR framework applies naturally.

Wu et al. (2024) imposes expert score function realizability and that the expected state norm remains bounded during learning. The algorithm has provable guarantees but it requires an expensive *RL in the loop* routine that we avoid in our work.

**Exploration Techniques in Deep RL** Ensemble of $Q$ networks has also been used for training stabilization (Anschel et al., 2017). (Zhang et al., 2025) introduces exploration technique based on multiple actors. Ciosek et al. (2019) does not have theoretical guarantees but it uses the idea of constructing an optimistic critic using mean plus standard deviation but only to define an exploratory policy with which collecting data. Our approach instead maintains only one actor policy which is updated with the optimistic $Q$ estimate. (Parker-Holder et al., 2020; Lyu et al., 2022) exploration with ensemble of actors rather than critics. (Kurutach et al., 2018; Chua et al., 2018) uses an ensemble of networks trained to learn the transition model to improve the sample complexity in model based RL. (Henaff et al., 2022) learns instead an inverse dynamics model

and via an encoder and decoder model and uses the features output by the encoder to compute elliptical potential bonuses which are standard in linear bandits (Abbasi-Yadkori et al., 2011). Moskovitz et al. (2021) improved TD-3 (Fujimoto et al., 2018) using an ensemble of critics and a bandit algorithm to find an aggregation rule balancing well the amount of optimism required by online exploration and pessimism required by off policy algorithms such as TD-3.

In addition, there are several deep RL work that takes a bayesian point of view to the problem, these algorithms often achieve remarkable performance but the algorithm implemented with deep networks requires usually adjustments creating a mismatch compared to the provable algorithms in the tabular case. Among those (Luis et al., 2023; Zhou et al., 2020; O'Donoghue et al., 2018) use the Bellman equation for the state value function variance to train a network (dubbed $U$ network) that models the uncertainty of the network predicting the $Q$ values. They respectively prove that this trick improves the performances of SAC, PPO (Schulman et al., 2017) and DQN(Mnih et al., 2015). (Curi et al., 2020) uses the model uncertainty estimate in the update of the actor.

Moreover, building on the theoretical analysis of PSRL (Osband & Van Roy, 2014) and RLSVI (Osband et al., 2016b) that show sublinear bayesian regret bound. At any step, these algorithms sample from a posterior distribution either an MDP where to plan or a value function to follow greedly at each step. Between one step and the other the posterior is updated given the new data. While the theorical analysis in the above works prescribe a randomization at the value function parameters level, in the deep RL version, dubbed Boostrapped DQN (Osband et al., 2016a), the perturbation is performed implicitly maintaining a set of $Q$ networks and sampling uniformly at each round according to which network the agent chooses the greedy action. (Chen et al., 2017) improved upon Bootstrapped DQN using an aggregation rule. That is acting greedy with respect to the mean plus standard deviation of the $q$ ensemble. Osband et al. (2018) further builds on this idea adding a differ prior to each network in the ensemble to increase diversity. Finally, Osband et al. (2023a) replaces the uniform sampling in (Osband et al., 2016a) with a learned distribution with an epistemic network (Osband et al., 2023b).

Furthermore, motivated by the bayesian regret bound proven in (O'Donoghue, 2021) in the tabular case and the one in (O'Donoghue, 2023), (Tarbouriech et al., 2024) proves a regret bound in the function approximation setting and showcased convincing performance in the Atari benchmark. Their algorithm requires to know the variance of the cost posterior distribution which is not available in the neural network experiments. Therefore, it is estimated using the standard deviation of an ensemble of cost network. In our work, we use an ensemble of $Q$ networks and not cost networks.

Additionally, (Ishfaq et al., 2021) analyzed ensemble exploration techniques in the general function approximation setting. Their ensemble consists of different critics trained on the same state actions dataset but with rewards perturbed with a gaussian random vector. (Ishfaq et al., 2023; 2024) looked at efficient implementation of Thompson sampling in Deep RL and obtained convincing results in Atari and providing guarantees for linear MDPs and general function approximation respectively. Moreover, (Ishfaq et al., 2025) extended the above results for continuous action spaces. Unfortunately, these methods do not apply directly to imitation learning because they require a fixed reward function.

**Exploration techniques in Deep IL**     As mentioned only few works investigated exploration techniques in Deep IL. Apart from the previously mentioned works, (Yu et al., 2020) adopts a model based approach and used exploration bonuses based on prediction error of the next observed state (a.k.a. curiosity driven exploration (Pathak et al., 2017; Burda et al., 2018)). Finally we notice that ensembles have been used in IL theory IL also for goals different to exploration. In particular, (Swamy et al., 2022) partitioned the expert dataset in two subdataset and show that these technique allows for improved expert sample complexity bounds when the expert is deterministic.

**State-only imitation learning**     Torabi et al. (2018b) tackled the problem of imitation learning from states only modifying the discriminator of GAIL (Ho & Ermon, 2016) to take as input state next state pairs instead of state action pairs. Further practical improvements have been proposed in (Zhu et al., 2020) that allows for the use of off-policy data. The works (Yang et al., 2019; Nair et al., 2017; Pathak et al., 2018; Radosavovic et al., 2021) use the idea of an inverse dynamic model while (Edwards et al., 2019; Ganai et al., 2023) develops a practical algorithm aiming at estimating the forward dynamic model. Furthermore, (Torabi et al., 2018a) introduces a twist in behavioral cloning using inverse dynamic modelling to make it applicable to state only expert datasets. A comprehensive literature review can be found in (Torabi et al., 2019). More recently, features/state only imitation learning has found application in non markovian decision making problems (Qin et al., 2024). Sikchi et al. (2022) introduce an algorithm that takes advantage of an offline ranker between trajectories to get strong empirical results in LfO setting. Another line of works (Gupta et al., 2017; Sermanet et al., 2018; Liu et al., 2019; Viano et al., 2021; 2022a; Gangwani & Peng, 2020; Cao & Sadigh, 2021; Gangwani et al., 2022) motivate imitation learning from observation alone arguing that the expert providing the demonstrations and the learner acts in slightly different environments.

In (Kim et al., 2022a; Sikchi et al., 2024), the authors proposed convex programming based methods to imitate an expert policy from expert state only demonstration and auxiliary arbitrary state action pairs. Several works (Ni et al., 2021; Kim et al., 2022b; Ma et al., 2022; Yu et al., 2023) introduce empirical methods to minimize an $f$-divergence between expert and learner state occupancy measure. Complementary, (Chang et al., 2023) minimizes the Wasserstein distance between expert and learner state occupancy measure. Their numerical results are convincing but no sample complexity bounds are provided. Convincing results have been obtained also in (Chang et al., 2024) that uses the idea of boosting and in (Wu et al., 2024) which uses a diffusion models inspired loss to update the cost.

## B. Theoretical Analysis

### B.1. Upper bounding the policy regret

**Corollary B.1.** *Consider an MDP where* $\max_{s,a \in \mathcal{S} \times \mathcal{A}} \mathrm{supp}(P(\cdot|s,a)) = 2$, *then for* $L \geq 36 \log \left( \frac{|\mathcal{S}||\mathcal{A}|K}{\delta} \right)$ *it holds that with probability at least* $1 - \delta$

$$\min_{\ell \in [L]} \widehat{P}_\ell^k V^k(s,a) \leq PV^k(s,a) \quad \forall \quad s, a \in \mathcal{S} \times \mathcal{A}, \quad \forall \quad k \in [K].$$

*Proof.* Let us fix a state-action-next state triplet $s, a, s'$, a batch index $\ell \in [L]$ and an iteration index $k \in [K]$. Then, we consider the following stochastic estimator for the probability of transitioning to $s'$ from state $s$ taking action $a$.

$$\widehat{P}_\ell^k(s'|s,a) = \frac{N_\ell^k(s,a,s')}{N_\ell^k(s,a)+2} = \frac{\sum_{\bar{s},\bar{a},\bar{s}' \in \mathcal{R}_\ell^k} \mathbb{1}_{\{\bar{s},\bar{a},\bar{s}'=s,a,s'\}}}{N_\ell^k(s,a)+2}$$

$$= \frac{\sum_{\bar{s},\bar{a},\bar{s}' \in \mathcal{R}_\ell^k : \bar{s},\bar{a}=s,a} \mathbb{1}_{\{\bar{s}'=s'\}}}{N_\ell^k(s,a)+2}.$$

Notice that in above estimators the denominator corresponds to the number of visits of the pair $s, a$ up to the time $k \in [K]$ within the batch $\ell \in [L]$, i.e. $N_\ell^k(s,a)$, increased by 2 for technical reasons. At the numerator instead we have the sum of $N_\ell^k(s,a)$ indicators functions which equals one when the state following the state action pair $s, a$ is equal to $s'$. Each of this indicator is a random variable distributed according to a Bernoulli random variable with mean $P(s'|s,a)$. At this point, we can use the technique introduced in (Cassel et al., 2024). In particular, we will show that

$$\mathbb{P}\left[ \widehat{P}_\ell^k(s'|s,a) \leq P(s'|s,a) \right] \geq \frac{1}{4} \quad \forall s, a, s' \in \mathcal{S} \times \mathcal{A} \times \mathrm{children}(s), \forall k \in [K], \forall \ell \in [L].$$

We distinguish 3 cases: $N_\ell^k(s,a) = 0$, $N_\ell^k(s,a) = 1$, $N_\ell^k(s,a) \geq 2$. If $N_\ell^k(s,a) = 0$, then, we have that

$$\widehat{P}_\ell^k(s'|s,a) = 0 \leq P(s'|s,a).$$

If $N_\ell^k(s,a) = 1$, we distinguish two cases. If $P(s'|s,a) \geq \frac{1}{3}$, then

$$\widehat{P}_\ell^k(s'|s,a) \leq \frac{1}{3} \leq P(s'|s,a).$$

Otherwise, if $N_\ell^k(s,a) = 1$, and $P(s'|s,a) \leq \frac{1}{3}$ then

$$\mathbb{P}\left[ \frac{\mathbb{1}_{\{\bar{s}'=s'\}}}{3} \leq P(s'|s,a) \right] = \mathbb{P}\left[ \mathbb{1}_{\{\bar{s}'=s'\}} = 0 \right] = 1 - P(s'|s,a) \geq \frac{2}{3} \geq \frac{1}{4}.$$

Finally, for $N_\ell^k(s,a) \geq 2$, we have that for $P(s'|s,a) \geq 1 - \frac{1}{N_\ell^k(s,a)}$ it holds that

$$\widehat{P}_\ell^k(s'|s,a) \leq \frac{N_\ell^k(s,a)}{N_\ell^k(s,a)+2} = 1 - \frac{2}{N_\ell^k(s,a)+2} \leq 1 - \frac{1}{N_\ell^k(s,a)} \leq P(s'|s,a),$$

where the last inequality holds for $N_\ell^k \geq 2$. Otherwise, for $P(s'|s,a) \leq 1 - \frac{1}{N_\ell^k(s,a)}$ we can apply (Cassel et al., 2024, Lemma 2) (adapted from (Wiklund, 2023, Corollary 1) ) to obtain

$$\mathbb{P}\left[ \widehat{P}_\ell^k(s'|s,a) \leq P(s'|s,a) \right] \geq \mathbb{P}\left[ \sum_{\bar{s},\bar{a},\bar{s}' \in \mathcal{R}_\ell^k : \bar{s},\bar{a}=s,a} \mathbb{1}_{\{\bar{s}'=s'\}} \leq N_\ell^k(s,a)P(s'|s,a) \right] \geq \frac{1}{4}.$$

At this point notice that for any positive vector $V \in [0, (1-\gamma)^{-1}]^{|\mathcal{S}|}$, it holds that

$$\mathbb{P}\left[\widehat{P}_\ell^k(s'|s,a) \leq P(s'|s,a) \ \forall s' \in \text{children}(s)\right] = \mathbb{P}\left[\widehat{P}_\ell^k(s'|s,a)V(s') \leq P(s'|s,a)V(s') \ \forall s' \in \text{children}(s)\right]$$

$$\leq \mathbb{P}\left[\sum_{s'\in\mathcal{S}} \widehat{P}_\ell^k(s'|s,a)V(s') \leq \sum_{s'\in\mathcal{S}} P(s'|s,a)V(s')\right]$$

where the inequality holds because of the following implication between events

$$\widehat{P}_\ell^k(s'|s,a)V(s') \leq P(s'|s,a)V(s') \ \forall s' \in \text{children}(s)$$

$$\implies \sum_{s'\in\text{children}(s)} \widehat{P}_\ell^k(s'|s,a)V(s') \leq \sum_{s'\in\text{children}(s)} P(s'|s,a)V(s').$$

Moreover, since the estimation at each state $s'$ is independent we have that

$$\mathbb{P}\left[\widehat{P}_\ell^k(s'|s,a) \leq P(s'|s,a) \ \forall s' \in \text{children}(s)\right] = \prod_{s'\in\text{children}(s)} \mathbb{P}\left[\widehat{P}_\ell^k(s'|s,a) \leq P(s'|s,a)\right] \geq \left(\frac{1}{4}\right)^{|\text{children}(s)|}$$

Then, for any $s, a \in \mathcal{S} \times \mathcal{A}$ is concluded by the following chain of inequalities

$$\mathbb{P}\left[\min_{\ell\in[L]} \sum_{s'\in\text{children}(s)} \widehat{P}_\ell^k(s'|s,a)V(s') \geq \sum_{s'\in\text{children}(s)} P(s'|s,a)V(s')\right]$$

$$= \mathbb{P}\left[\sum_{s'\in\text{children}(s)} \widehat{P}_\ell^k(s'|s,a)V(s') \geq \sum_{s'\in\text{children}(s)} P(s'|s,a)V(s') \ \forall\ell \in [L]\right]$$

$$= \prod_{\ell\in[L]}\left(1 - \mathbb{P}\left[\sum_{s'\in\text{children}(s)} \widehat{P}_\ell^k(s'|s,a)V(s') \leq \sum_{s'\in\text{children}(s)} P(s'|s,a)V(s')\right]\right)$$

$$\leq \prod_{\ell\in[L]}\left(1 - \mathbb{P}\left[\widehat{P}_\ell^k(s'|s,a) \leq P(s'|s,a) \ \forall s' \in \text{children}(s)\right]\right)$$

$$\leq \prod_{\ell\in[L]}\left(1 - \frac{1}{4}^{|\text{children}(s)|}\right) \leq e^{-L\log\left(\frac{4^{|\text{children}(s)|}}{4^{|\text{children}(s)|}-1}\right)}.$$

Therefore, choosing $L \geq \frac{\log(1/\delta)}{\log\left(\frac{4^{|\text{children}(s)|}}{4^{|\text{children}(s)|}-1}\right)}$, we ensure that

$$\mathbb{P}\left[\min_{\ell\in[L]} \widehat{P}_\ell^k V^k(s,a) \geq PV^k(s,a)\right] \leq \delta.$$

For $|\text{children(s)}| = 2$, we have that $\left(\log\left(\frac{4^{|\text{children}(s)|}}{4^{|\text{children}(s)|}-1}\right)\right)^{-1} \leq 36$, therefore with a union bound over the sets $\mathcal{S}, \mathcal{A}$ and $[K]$ we conclude that for $L \geq 36\log\left(\frac{|\mathcal{S}||\mathcal{A}|K}{\delta}\right)$, it holds that

$$\mathbb{P}\left[\min_{\ell\in[L]} \widehat{P}_\ell^k V^k(s,a) \leq PV^k(s,a) \ \forall\ s,a,k \in \mathcal{S} \times \mathcal{A} \times [K]\right] \geq 1 - \delta.$$

$\square$

## B.2. Policy Regret Decomposition

**Theorem 4.2.** *Policy Regret In a binarized MDP with $|\mathcal{S}|$ states and discount factor $\gamma$, it holds that with probability $1 - 3\delta$, for any policy $\pi^\star$, $\text{Regret}_\pi(K, \pi^\star)$ is upper bounded by*

$$\frac{\log|\mathcal{A}|}{\eta(1-\gamma)} + \frac{\eta K}{(1-\gamma)^4} + \widetilde{\mathcal{O}}\left(\frac{\sqrt{K|\mathcal{S}|^2|\mathcal{A}|\log(1/\delta)}}{(1-\gamma)^2}\right)$$

*and for $\eta = \sqrt{\frac{\log|\mathcal{A}|(1-\gamma)^3}{K}}$ it holds that using the update in* (Min) *or in* (Mean-Std) *it holds that* $\mathrm{Regret}(K, \pi^\star)$ *is upper bounded by* $\widetilde{\mathcal{O}}\left(\sqrt{\frac{K|\mathcal{S}|^2|\mathcal{A}|\log(1/\delta)}{(1-\gamma)^5}}\right)$.

*Proof.* The theorem is proven with the following regret decomposition in virtue of Lemma 4.5 already presented in the main text. Denoting $\delta^k(s, a) \triangleq c^k(s, a) + \gamma P V^k(s, a) - Q^{k+1}(s, a)$ and $g^k(s, a) \triangleq Q^{k+1}(s, a) - Q^k(s, a)$

$$(1 - \gamma)\mathrm{Regret}_\pi(K; \pi^\star) = \sum_{k=1}^{K} \mathbb{E}_{s \sim d^{\pi^\star}}\left[\langle Q^k(s, \cdot), \pi^k(s) - \pi^\star(s)\rangle\right] \tag{BTRL}$$

$$+ \sum_{k=1}^{K} \sum_{s,a} \left[d^{\pi^k}(s, a) - d^{\pi^\star}(s, a)\right] \cdot \left[\delta^k(s, a)\right] \tag{Optimism}$$

$$+ \sum_{k=1}^{K} \mathbb{E}_{s,a \sim d^{\pi^k}}\left[g^k(s, a)\right] - \sum_{k=1}^{K} \mathbb{E}_{s,a \sim d^{\pi^\star}}\left[g^k(s, a)\right] \tag{Shift}$$

At this point, we bound each term with the following Lemmas, we obtain

$$\sqrt{\frac{\log|\mathcal{A}| K}{(1-\gamma)^5}} + \widetilde{\mathcal{O}}\left(\frac{\sqrt{K |\mathcal{S}|^2 |\mathcal{A}| \log(1/\delta)}}{(1-\gamma)^2}\right)$$

$$\leq \widetilde{\mathcal{O}}\left(\sqrt{\frac{K |\mathcal{S}|^2 |\mathcal{A}| \log(1/\delta)}{(1-\gamma)^5}}\right).$$

$\square$

**Bound on** (BTRL) Then, we continue bounding the first term invoking the following lemma.

**Lemma B.2.** *Let us consider the sequence of policies $\left\{\pi^k\right\}_{k=1}^{K}$ generated by Algorithm 2 for all $\eta > 0$ then it holds that* (BTRL) $\leq \frac{\log|\mathcal{A}|}{\eta}$.

*Proof.*

$$\sum_{k=1}^{K} \mathbb{E}_{s \sim d^{\pi^\star}_{\rho_{e_k}}}\left[\langle Q^k(s, \cdot), \pi^k(s) - \pi^\star(s)\rangle\right] = \sum_{k \in K} \mathbb{E}_{s \sim d^{\pi^\star}}\left[\langle Q^k(s, \cdot), \pi^k(s) - \pi^\star(s)\rangle\right]$$

$$= \mathbb{E}_{s \sim d^{\pi^\star}}\left[\sum_{k \in K} \langle Q^k(s, \cdot), \pi^k(s) - \pi^\star(s)\rangle\right] \tag{2}$$

Then, applying a standard regret bound for Be the Regularized Leader (BTRL) it holds that for all $s \in \mathcal{S}$

$$\sum_{k \in K} \langle Q^k(s, \cdot), \pi^k(s) - \pi^\star(s)\rangle = \frac{\log|\mathcal{A}|}{\eta}$$

Then, plugging into (2) we conclude that

$$(\text{BTRL}) \leq \frac{\log|\mathcal{A}|}{\eta}.$$

$\square$

**Bound on** (Shift) We follow the idea from (Moulin & Neu, 2023) of controlling this term proving that two consecutive policies will have occupancy measures within a $\mathcal{O}(\eta)$ total variation distance.

**Lemma B.3.** *For the sequence of policies $\left\{\pi^k\right\}_{k=1}^{K}$ generated by Algorithm 2, for all $\eta > 0$, it holds that* (Shift) $\leq \frac{\eta K}{(1-\gamma)^3}$.

*Proof.* Let us denote $Q_{\max} = (1-\gamma)^{-1}$

$$\sum_{k=1}^{K} \left\langle d^{\pi^k}, Q^k - Q^{k+1} \right\rangle = \left( \left\langle d^{\pi^1}, Q^1 \right\rangle + \sum_{2}^{K-1} \left\langle Q^k, d^{\pi^k} - d^{\pi^{k-1}} \right\rangle - \left\langle d^{\pi^K}, Q^K \right\rangle \right)$$

$$\leq Q_{\max} + \frac{\eta Q_{\max}^2 (K-2)}{(1-\gamma)} \leq \frac{\eta Q_{\max}^2 (K-1)}{(1-\gamma)}.$$

In the first inequality, we used Lemma C.4. Finally, noticing that

$$-\sum_{k=1}^{K} \left\langle d^{\pi^\star}, Q^k - Q^{k+1} \right\rangle = \left\langle d^{\pi^\star}, Q^K - Q^1 \right\rangle \leq Q_{\max}$$

and that

$$(\text{Shift}) = \sum_{k=1}^{K} \left\langle d^{\pi^k}, Q^k - Q^{k+1} \right\rangle - \sum_{k=1}^{K} \left\langle d^{\pi^\star}, Q^k - Q^{k+1} \right\rangle$$

allows us to conclude the proof summing the two bounds. $\square$

**Bound on** (Optimism)

**Lemma B.4.** *Let us consider an MDP where $\max_{s,a \in \mathcal{S} \times \mathcal{A}} \operatorname{supp}(P(\cdot|s,a)) = 2$. For each $k \in [K]$, if the $Q^{k+1}$ in Algorithm 2, are updated according to* (Min) *or* (Mean-Std)*, the iterates produced by Algorithm 2 satisfy with probability $1 - 3\delta$ that*

$$(\text{Optimism}) \leq \widetilde{\mathcal{O}} \left( \frac{\sqrt{K |\mathcal{S}|^2 |\mathcal{A}| \log(1/\delta)}}{1-\gamma} \right).$$

*Proof.* For the optimism term we can observe that using the update of the $Q$ values, we have that

$$\delta^k(s,a) = \gamma(PV^k(s,a) - \min_{\ell \in [L]} \widehat{P}_\ell^k V^k(s,a))$$

Therefore, in virtue of Corollary 4.11 with probability $1 - \delta$ it holds that

$$\delta^k(s,a) \geq 0 \quad \forall \quad s,a \in \mathcal{S} \times \mathcal{A}.$$

Therefore, with probability $1 - \delta$, we have that

$$\sum_{k=1}^{K} \sum_{s,a} \left[ d^{\pi^k}(s,a) - d^{\pi^\star}(s,a) \right] \cdot \left[ \delta^k(s,a) \right] \leq \sum_{k=1}^{K} \sum_{s,a} d^{\pi^k}(s,a) \delta_k(s,a)$$

$$= \gamma \sum_{k=1}^{K} \mathbb{E}_{s,a \sim d^{\pi^k}} \left[ PV^k(s,a) - \min_{\ell \in [L]} \widehat{P}_\ell^k V^k(s,a) \right]$$

At this point, using Lemma B.5 and a union bound we have that with probability $1 - 2\delta$, it holds that

$$\sum_{k=1}^{K} \sum_{s,a} \left[ d^{\pi^k}(s,a) - d^{\pi^\star}(s,a) \right] \cdot \left[ \delta^k(s,a) \right]$$

$$\leq \gamma \sum_{k=1}^{K} \mathbb{E}_{s,a \sim d^{\pi^k}} \left[ \sqrt{\frac{|\mathcal{S}|}{(N^k(s,a)/L + 1)(1-\gamma)^2} \log \left( \frac{|\mathcal{S}| |\mathcal{A}| LK^2(K+1)}{(1-\gamma)\delta} \right)} \right]$$

$$+ \gamma \sum_{k=1}^{K} \mathbb{E}_{s,a \sim d^{\pi^k}} \left[ \frac{2}{(N^k(s,a)/L + 1)(1-\gamma)} \right] + 4$$

$$\leq \sqrt{K \sum_{k=1}^{K} \mathbb{E}_{s,a \sim d^{\pi^k}} \left[ \frac{|\mathcal{S}|}{(N^k(s,a)/L + 1)(1-\gamma)^2} \log \left( \frac{|\mathcal{S}| |\mathcal{A}| L K^2 (K+1)}{(1-\gamma)\delta} \right) \right]}$$

$$+ \gamma \sum_{k=1}^{K} \mathbb{E}_{s,a \sim d^{\pi^k}} \left[ \frac{2}{(N^k(s,a)/L + 1)(1-\gamma)} \right] + 4$$

At this point, since $L^k$ is geometrically distributed for every $k \in [K]$ it holds that $L^k \leq L_{\max} := \frac{\log(K/\delta)}{(1-\gamma)}$ for all $k \in [K]$ with probability $1 - \delta$. Therefore, we can invoke Lemma C.2 to bound $\sum_{k=1}^{K} \mathbb{E}_{s,a \sim d^{\pi^k}} \left[ \frac{2}{(N^k(s,a)/L+1)} \right]$. Another union bound ensures that with probability $1 - 3\delta$, we have that

$$\sum_{k=1}^{K} \sum_{s,a} \left[ d^{\pi^k}(s,a) - d^{\pi^\star}(s,a) \right] \cdot \left[ \delta^k(s,a) \right]$$

$$\leq \frac{1}{1-\gamma} \sqrt{K |\mathcal{S}| \log \left( \frac{|\mathcal{S}| |\mathcal{A}| L K^2 (K+1)}{(1-\gamma)\delta} \right) \left( 2L |\mathcal{S}| |\mathcal{A}| \log (K L_{\max}) + 4 \log (2K/\delta) \right)}$$

$$+ \frac{2}{1-\gamma} \left( 2L |\mathcal{S}| |\mathcal{A}| \log (K L_{\max}) + 4 \log (2K/\delta) \right)$$

$$= \widetilde{\mathcal{O}} \left( \frac{\sqrt{K |\mathcal{S}|^2 |\mathcal{A}| \log(1/\delta)}}{1-\gamma} \right).$$

where the $\widetilde{\mathcal{O}}$ notation hides logarithmic factors in $K, 1-\gamma, |\mathcal{S}|$ and $|\mathcal{A}|$.

Now, we prove the part of the Theorem that considers the update for $Q^{k+1}$ given in (Mean-Std). Under this update, we have that

$$\delta^k(s,a) = \gamma \left( PV^k(s,a) - \max \left[ \frac{1}{L} \sum_{\ell=1}^{L} \widehat{P}_\ell^k V^k(s,a) - \sqrt{\sum_{\ell=1}^{L} \left( \widehat{P}_\ell^k V^k(s,a) - \frac{1}{L} \sum_{\ell=1}^{L} \widehat{P}_\ell^k V^k(s,a) \right)^2}, 0 \right] \right)$$

Then, applying Samuelson's inequality Lemma C.5, we have that

$$\min_{\ell \in [L]} \widehat{P}_\ell^k V^k(s,a) \geq \frac{1}{L} \sum_{\ell=1}^{L} \widehat{P}_\ell^k V^k(s,a) - \sqrt{\sum_{\ell=1}^{L} \left( \widehat{P}_\ell^k V^k(s,a) - \frac{1}{L} \sum_{\ell=1}^{L} \widehat{P}_\ell^k V^k(s,a) \right)^2}$$

moreover, it holds that

$$\min_{\ell \in [L]} \widehat{P}_\ell^k V^k(s,a) \geq 0$$

Therefore,

$$\min_{\ell \in [L]} \widehat{P}_\ell^k V^k(s,a) \geq \max \left[ \frac{1}{L} \sum_{\ell=1}^{L} \widehat{P}_\ell^k V^k(s,a) - \sqrt{\sum_{\ell=1}^{L} \left( \widehat{P}_\ell^k V^k(s,a) - \frac{1}{L} \sum_{\ell=1}^{L} \widehat{P}_\ell^k V^k(s,a) \right)^2}, 0 \right]$$

which implies

$$\delta^k(s,a) \geq \gamma \left( PV^k(s,a) - \min_{\ell \in [L]} \widehat{P}_\ell^k V^k(s,a) \right) \geq 0$$

where the last inequality holds with probability $1 - \delta$ thanks to Corollary 4.11. Therefore, with probability $1 - \delta$, we have that

$$\sum_{k=1}^{K} \sum_{s,a} \left[ d^{\pi^k}(s,a) - d^{\pi^\star}(s,a) \right] \cdot \left[ \delta^k(s,a) \right] \leq \sum_{k=1}^{K} \sum_{s,a} d^{\pi^k}(s,a) \delta_k(s,a)$$

$$\leq \gamma \sum_{k=1}^{K} \mathbb{E}_{s,a \sim d^{\pi^k}} \left[ \frac{1}{L} \sum_{\ell=1}^{L} \left( P V^k(s,a) - \widehat{P}_\ell^k V^k(s,a) \right) + \sqrt{\sum_{\ell=1}^{L} \left( \widehat{P}_\ell^k V^k(s,a) - \frac{1}{L} \sum_{\ell=1}^{L} \widehat{P}_\ell^k V^k(s,a) \right)^2} \right],$$

where the last inequality holds removing the maximum. For the first term, we can use Lemma B.5 and continue as in the previous proof to show that with probability $1 - \delta$

$$\gamma \sum_{k=1}^{K} \mathbb{E}_{s,a \sim d^{\pi^k}} \left[ \frac{1}{L} \sum_{\ell=1}^{L} \left( P V^k(s,a) - \widehat{P}_\ell^k V^k(s,a) \right) \right] \leq \widetilde{\mathcal{O}} \left( \frac{\sqrt{K |\mathcal{S}|^2 |\mathcal{A}| \log(1/\delta)}}{1 - \gamma} \right).$$

So, we are left with bounding the term

$$\gamma \sum_{k=1}^{K} \mathbb{E}_{s,a \sim d^{\pi^k}} \left[ \sqrt{\sum_{\ell=1}^{L} \left( \widehat{P}_\ell^k V^k(s,a) - \frac{1}{L} \sum_{\ell'=1}^{L} \widehat{P}_{\ell'}^k V^k(s,a) \right)^2} \right]$$

To this end, by Jensen's inequality we have that

$$\gamma \sum_{k=1}^{K} \mathbb{E}_{s,a \sim d^{\pi^k}} \left[ \sqrt{\sum_{\ell=1}^{L} \left( \widehat{P}_\ell^k V^k(s,a) - \frac{1}{L} \sum_{\ell'=1}^{L} \widehat{P}_{\ell'}^k V^k(s,a) \right)^2} \right]$$

$$\leq \gamma \sum_{k=1}^{K} \mathbb{E}_{s,a \sim d^{\pi^k}} \left[ \sqrt{\frac{1}{L} \sum_{\ell'=1}^{L} \sum_{\ell=1}^{L} \left( \widehat{P}_\ell^k V^k(s,a) - \widehat{P}_{\ell'}^k V^k(s,a) \right)^2} \right]$$

$$\leq \gamma \sum_{k=1}^{K} \mathbb{E}_{s,a \sim d^{\pi^k}} \left[ \sqrt{\frac{2}{L} \sum_{\ell'=1}^{L} \sum_{\ell=1}^{L} \left( \widehat{P}_\ell^k V^k(s,a) - P V^k(s,a) \right)^2} \right]$$

$$\leq \gamma \sum_{k=1}^{K} \mathbb{E}_{s,a \sim d^{\pi^k}} \left[ \sqrt{2 \sum_{\ell=1}^{L} \left( \widehat{P}_\ell^k V^k(s,a) - P V^k(s,a) \right)^2} \right]$$

At this point, notice that invoking Lemma B.5, we have that for all $\ell \in [L]$ with probability $1 - \delta$

$$\left( \widehat{P}_\ell^k V^k(s,a) - P V^k(s,a) \right)^2$$

$$\leq \left( \sqrt{\frac{|\mathcal{S}|}{(N^k(s,a)/L + 1)(1 - \gamma)^2} \log \left( \frac{|\mathcal{S}| |\mathcal{A}| L K^2 (K+1)}{(1 - \gamma) \delta} \right)} + \frac{2}{K} + \frac{2}{(N^k(s,a)/L + 1)(1 - \gamma)} \right)^2$$

$$\leq \frac{3 |\mathcal{S}|}{(N^k(s,a)/L + 1)(1 - \gamma)^2} \log \left( \frac{|\mathcal{S}| |\mathcal{A}| L K^2 (K+1)}{(1 - \gamma) \delta} \right) + \frac{12}{K^2} + \frac{12}{(N^k(s,a)/L + 1)^2 (1 - \gamma)^2}$$

$$= \widetilde{\mathcal{O}} \left( \frac{3 |\mathcal{S}| \log \left( \frac{1}{\delta} \right)}{(N^k(s,a)/L + 1)(1 - \gamma)^2} \right).$$

Therefore, plugging into the previous display we obtain

$$\gamma \sum_{k=1}^{K} \mathbb{E}_{s,a \sim d^{\pi^k}} \left[ \sqrt{\sum_{\ell=1}^{L} \left( \widehat{P}_\ell^k V^k(s,a) - \frac{1}{L} \sum_{\ell'=1}^{L} \widehat{P}_{\ell'}^k V^k(s,a) \right)^2} \right]$$

$$\leq \sum_{k=1}^{K} \mathbb{E}_{s,a\sim d^{\pi^k}} \left[ \sqrt{\sum_{\ell=1}^{L} \widetilde{\mathcal{O}} \left( \frac{3\,|\mathcal{S}|\log\left(\frac{1}{\delta}\right)}{(N^k(s,a)/L + 1)(1-\gamma)^2} \right)} \right]$$

$$\leq \sqrt{K \sum_{k=1}^{K} \mathbb{E}_{s,a\sim d^{\pi^k}} \left[ \sum_{\ell=1}^{L} \widetilde{\mathcal{O}} \left( \frac{3\,|\mathcal{S}|\log\left(\frac{1}{\delta}\right)}{(N^k(s,a)/L + 1)(1-\gamma)^2} \right) \right]}$$

$$\leq \widetilde{\mathcal{O}} \left( \sqrt{K \sum_{k=1}^{K} \mathbb{E}_{s,a\sim d^{\pi^k}} \left[ \frac{3\,|\mathcal{S}|\,L^2\log\left(\frac{1}{\delta}\right)}{(N^k(s,a) + 1)(1-\gamma)^2} \right]} \right)$$

$$\leq \widetilde{\mathcal{O}} \left( \sqrt{\frac{|\mathcal{S}|^2 \log\left(\frac{1}{\delta}\right) K}{(1-\gamma)^2} \sum_{k=1}^{K} \mathbb{E}_{s,a\sim d^{\pi^k}} \left[ \frac{1}{N^k(s,a) + 1} \right]} \right)$$

Finally, we bound $\mathbb{E}_{s,a\sim d^{\pi^k}} \left[ \frac{1}{N^k(s,a)+1} \right]$ using Lemma C.2 under the event $L^k \leq L_{\max} := \frac{\log(K/\delta)}{(1-\gamma)}$ which holds with probability $1 - \delta$. Thanks to an union bound we have that with probability $1 - 2\delta$,

$$\gamma \sum_{k=1}^{K} \mathbb{E}_{s,a\sim d^{\pi^k}} \left[ \sqrt{\sum_{\ell=1}^{L} \left( \widehat{P}_\ell^k V^k(s,a) - \frac{1}{L}\sum_{\ell'=1}^{L} \widehat{P}_{\ell'}^k V^k(s,a) \right)^2} \right]$$

$$\leq \widetilde{\mathcal{O}} \left( \sqrt{\frac{|\mathcal{S}|^2 \log\left(\frac{1}{\delta}\right) K}{(1-\gamma)^2} \, |\mathcal{S}|\,|\mathcal{A}|\log\left(KL_{\max}\right) + 4\log\left(2KL_{\max}/\delta\right)} \right)$$

$$\leq \widetilde{\mathcal{O}} \left( \frac{\sqrt{K\,|\mathcal{S}|^2\,|\mathcal{A}|\log(1/\delta)}}{1-\gamma} \right).$$

Therefore, the proof is concluded also for the case of $Q$ value being updated as in (Mean-Std). $\qquad\square$

**Lemma B.5.** *With probability $1 - \delta$, it holds that for all $s,a \in \mathcal{S} \times \mathcal{A}$,*

$$PV^k(s,a) - \widehat{P}_\ell^k V^k(s,a) \leq \sqrt{\frac{|\mathcal{S}|}{(N^k(s,a)/L + 1)(1-\gamma)^2} \log\left( \frac{|\mathcal{S}|\,|\mathcal{A}|\,LK^2(K+1)}{(1-\gamma)\delta} \right)} + \frac{2}{K}$$

$$+ \frac{2}{(N^k(s,a)/L + 1)(1-\gamma)} \quad \forall \ell, k \in [L] \times [K]$$

*In particular, the above statement implies that for all $k \in [K]$*

$$PV^k(s,a) - \min_{\ell\in[L]} \widehat{P}_\ell^k V^k(s,a) \leq \sqrt{\frac{|\mathcal{S}|}{(N^k(s,a)/L + 1)(1-\gamma)^2} \log\left( \frac{|\mathcal{S}|\,|\mathcal{A}|\,LK^2(K+1)}{(1-\gamma)\delta} \right)} + \frac{2}{K}$$

$$+ \frac{2}{(N^k(s,a)/L + 1)(1-\gamma)}$$

*Proof.* Let us introduce the value class of the possible value functions generated by Algorithm 2. i.e. $\mathcal{V} = \left\{ f \in \mathbb{R}^{|S|} \mid \|f\|_\infty \leq \frac{1}{1-\gamma}, f(s) \geq 0 \ \forall s \in \mathcal{S} \right\}$. Let us introduce a $\epsilon_{\text{cov}}$-covering set $\mathcal{C}_{\epsilon_{\text{cov}}}(\mathcal{V})$ such that for any $V \in \mathcal{V}$ there exists $\tilde{V} \in \mathcal{C}_{\epsilon_{\text{cov}}}(\mathcal{V})$ such that $\left\| \tilde{V} - V \right\|_\infty \leq \epsilon_{\text{cov}}$. Therefore let us denote by $\tilde{V}^k$ the element of $\mathcal{C}_{\epsilon_{\text{cov}}}(\mathcal{V})$ such that $\left\| V^k - \tilde{V}^k \right\|_\infty \leq \epsilon_{\text{cov}}$. Then, let us consider a generic $\tilde{V} \in \mathcal{C}_{\epsilon_{\text{cov}}}(\mathcal{V})$,

$$P\tilde{V}(s,a) - \frac{1}{N_\ell^k(s,a)} \sum_{\bar{s},\bar{a},s'\in\mathcal{R}_\ell^k} \tilde{V}(s')\mathbb{1}_{\{s,a=\bar{s},\bar{a}\}} = \frac{1}{N_\ell^k(s,a)} \sum_{\bar{s},\bar{a},s'\in\mathcal{R}_\ell^k} P\tilde{V}(s,a)\mathbb{1}_{\{s,a=\bar{s},\bar{a}\}}$$

$$- \frac{1}{N_\ell^k(s,a)} \sum_{\bar{s},\bar{a},s'\in\mathcal{R}_\ell^k} \tilde{V}(s') \mathbb{1}_{\{s,a=\bar{s},\bar{a}\}}$$

$$= \frac{1}{N_\ell^k(s,a)} \sum_{\bar{s},\bar{a},s'\in\mathcal{R}_\ell^k} \left( P\tilde{V}(s,a) - \tilde{V}(s') \right) \mathbb{1}_{\{s,a=\bar{s},\bar{a}\}}$$

Then, notice that denoting $s'_n(s,a)$ the state sample after $s,a$ the $n^{th}$ time the state action pair was visited we have that

$$\sum_{\bar{s},\bar{a},s'\in\mathcal{R}_\ell^k} \left( P\tilde{V}(s,a) - \tilde{V}(s') \right) \mathbb{1}_{\{s,a=\bar{s},\bar{a}\}} = \sum_{n=1}^{N_\ell^k(s,a)} \left( P\tilde{V}(s,a) - \tilde{V}(s'_n(s,a)) \right)$$

Applying directly the Azuma Hoeffding inequality is not possible because the number of elements in the sum, i.e. the number of visits $N_\ell^k(s,a)$ is not a random variable independent on the random variables $\{s'_n(s,a)\}_{n=1}^{N_\ell^k(s,a)}$ (see (Lattimore & Szepesvári, 2020, Exercise 7.1) ).

Therefore, we first apply the Azuma Hoeffding inequality for a specific $k$ and for a specific value of the visits $N_\ell^k(s,a)$. That is, it holds that with probability $1-\delta$

$$\sum_{\bar{s},\bar{a},s'\in\mathcal{R}_\ell^k} \left( P\tilde{V}(s,a) - \tilde{V}(s') \right) \mathbb{1}_{\{s,a=\bar{s},\bar{a}\}} \leq \sqrt{\frac{N_\ell^k(s,a)\log(1/\delta)}{2(1-\gamma)^2}}$$

Therefore via a union bound for $k\in[K]$ and $N_\ell^k(s,a)\in\{0,1,\dots,K\}$ we have that with probability $1-\delta$ it holds that for all $k\in[K]$

$$\sum_{\bar{s},\bar{a},s'\in\mathcal{R}_\ell^k} \left( P\tilde{V}(s,a) - \tilde{V}(s') \right) \mathbb{1}_{\{s,a=\bar{s},\bar{a}\}} \leq \sqrt{\frac{N_\ell^k(s,a)\log(K(K+1)/\delta)}{2(1-\gamma)^2}}.$$

Therefore, we can conclude that with probability at least $1-\delta$ for all $k\in[K]$

$$P\tilde{V}(s,a) - \frac{1}{N_\ell^k(s,a)} \sum_{\bar{s},\bar{a},s'\in\mathcal{R}_\ell^k} \tilde{V}(s')\mathbb{1}_{\{s,a=\bar{s},\bar{a}\}} \leq \sqrt{\frac{\log(K(K+1)/\delta)}{2N_\ell^k(s,a)(1-\gamma)^2}}.$$

Now, by a another union bound over $\mathcal{C}_{\epsilon_{\mathrm{cov}}}(\mathcal{V})$, $[K]$, $[L]$ and $\mathcal{S}\times\mathcal{A}$ and denoting

$$\bar{P}_\ell^k\tilde{V}(s,a) := \frac{1}{N_\ell^k(s,a)} \sum_{\bar{s},\bar{a},s'\in\mathcal{R}_\ell^k} \tilde{V}(s')\mathbb{1}_{\{s,a=\bar{s},\bar{a}\}},$$

it holds that

$$\mathbb{P}\left[ P\tilde{V}(s,a) - \bar{P}_\ell^k\tilde{V}(s,a) \leq \sqrt{\frac{\log(K(K+1)\,|\mathcal{S}|\,|\mathcal{A}|\,|\mathcal{C}_{\epsilon_{\mathrm{cov}}}(\mathcal{V})|\,L/\delta)}{N_\ell^k(s,a)(1-\gamma)^2}} \ \ \forall s,a,\ell,k\in\mathcal{S}\times\mathcal{A}\times[L]\times[K],\ \ \tilde{V}\in\mathcal{V} \right] \geq 1-\delta$$

Therefore, now let us consider the element $\tilde{V}^k\in\mathcal{C}_{\epsilon_{\mathrm{cov}}}(\mathcal{V})$ such that $\left\| V^k - \tilde{V}^k \right\|_\infty \leq \epsilon_{\mathrm{cov}}$. Then, we have that for all $\ell\in[L]$

$$PV^k(s,a) - \bar{P}_\ell^k V^k(s,a) = P\tilde{V}^k(s,a) - \bar{P}_\ell^k\tilde{V}^k(s,a) + (P - \bar{P}_\ell^k)(\tilde{V}^k - V^k)$$

$$= P\tilde{V}^k(s,a) - \bar{P}_\ell^k\tilde{V}^k(s,a) + 2\epsilon_{\mathrm{cov}}$$

$$\leq \sqrt{\frac{\log(K(K+1)\,|\mathcal{S}|\,|\mathcal{A}|\,|\mathcal{C}_{\epsilon_{\mathrm{cov}}}(\mathcal{V})|\,L/\delta)}{N_\ell^k(s,a)(1-\gamma)^2}} + 2\epsilon_{\mathrm{cov}}$$

$$\leq \sqrt{\frac{|\mathcal{S}|}{N_\ell^k(s,a)(1-\gamma)^2} \log\left(\frac{K(K+1)|\mathcal{S}||\mathcal{A}|L}{(1-\gamma)\epsilon_{\mathrm{cov}}\delta}\right)} + 2\epsilon_{\mathrm{cov}}$$

With $\epsilon_{\mathrm{cov}} = K^{-1}$, we get

$$PV^k(s,a) - \bar{P}_\ell^k V^k(s,a) \leq \sqrt{\frac{|\mathcal{S}|}{N_\ell^k(s,a)(1-\gamma)^2} \log\left(\frac{|\mathcal{S}||\mathcal{A}|LK^2(K+1)}{(1-\gamma)\delta}\right)} + \frac{2}{K}$$

Then, we can continue as follows

$$\widehat{P}_\ell^k V^k(s,a) = \frac{N_\ell^k(s,a)}{N_\ell^k(s,a)+2} \bar{P}_\ell^k V^k(s,a)$$

$$\geq \frac{N_\ell^k(s,a)}{N_\ell^k(s,a)+2} \left[ PV^k(s,a) - \sqrt{\frac{|\mathcal{S}|}{N_\ell^k(s,a)(1-\gamma)^2} \log\left(\frac{|\mathcal{S}||\mathcal{A}|LK^2(K+1)}{(1-\gamma)\delta}\right)} - \frac{2}{K} \right]$$

$$= PV^k(s,a) - \frac{2}{N_\ell^k(s,a)+2} PV^k(s,a) - \sqrt{\frac{|\mathcal{S}|}{(N_\ell^k(s,a)+2)(1-\gamma)^2} \log\left(\frac{|\mathcal{S}||\mathcal{A}|LK^2(K+1)}{(1-\gamma)\delta}\right)} - \frac{2}{K}$$

$$\geq PV^k(s,a) - \frac{2}{(N_\ell^k(s,a)+2)(1-\gamma)} - \sqrt{\frac{|\mathcal{S}|}{(N_\ell^k(s,a)+2)(1-\gamma)^2} \log\left(\frac{|\mathcal{S}||\mathcal{A}|LK^2(K+1)}{(1-\gamma)\delta}\right)} - \frac{2}{K}$$

Finally, rearranging and using that $N_\ell^k(s,a) = \left\lfloor \frac{N^k(s,a)}{L} \right\rfloor \geq \frac{N^k(s,a)}{L} - 1$, we obtain that with probability $1-\delta$, it holds that for all $\ell \in [L], k \in [K], s,a \in \mathcal{S} \times \mathcal{A}$

$$PV^k(s,a) - \widehat{P}_\ell^k V^k(s,a) \leq \sqrt{\frac{|\mathcal{S}|}{(N^k(s,a)/L+1)(1-\gamma)^2} \log\left(\frac{|\mathcal{S}||\mathcal{A}|LK^2(K+1)}{(1-\gamma)\delta}\right)} + \frac{2}{K}$$
$$+ \frac{2}{(N^k(s,a)/L+1)(1-\gamma)}$$

$\square$

## B.3. Upper bound the regret of the reward player

**Theorem 4.3.** *Cost Regret In a binarized MDP with $|\mathcal{S}|$ states and discount factor $\gamma$, it holds that with probability $1 - 2\delta$, $(1-\gamma)\mathrm{Regret}_c(K; c_{\mathrm{true}})$ is upper bounded by*

$$4\sqrt{K\log(1/\delta)} + K\sqrt{\frac{|\mathcal{S}||\mathcal{A}|\log(|\mathcal{S}||\mathcal{A}|/\delta)}{2|\mathcal{D}_{\pi_{\mathrm{E}}}|}}$$

*Proof.* We decompose the regret as follows

$$\sum_{k=1}^K \left\langle c_{\mathrm{true}} - c^k, d^{\pi^k} - d^{\pi_{\mathrm{E}}} \right\rangle = \sum_{k=1}^K \left\langle c_{\mathrm{true}} - c^k, \mathbf{e}_{s_{L^k}^k} - \widehat{d^{\pi_{\mathrm{E}}}} \right\rangle$$
$$+ \sum_{k=1}^K \left\langle c_{\mathrm{true}} - c^k, d^{\pi^k} - \mathbf{e}_{s_{L^k}^k} \right\rangle$$
$$+ \sum_{k=1}^K \left\langle c_{\mathrm{true}} - c^k, \widehat{d^{\pi_{\mathrm{E}}}} - d^{\pi_{\mathrm{E}}} \right\rangle$$

For the first term, we can invoke a standard online gradient descent bound and get

$$\sum_{k=1}^K \left\langle c_{\mathrm{true}} - c^k, \mathbf{e}_{s_{L^k}^k} - \widehat{d^{\pi_{\mathrm{E}}}} \right\rangle \leq \frac{2}{\eta} + \eta K \left\| \widehat{d^{\pi_{\mathrm{E}}}} \right\|_2 /2$$

$$\leq \frac{2}{\eta} + \eta K \left\| \widehat{d^{\pi_\mathrm{E}}} \right\|_1 / 2$$
$$\leq \frac{2}{\eta} + \frac{\eta K}{2}$$

Therefore choosing $\eta = \sqrt{\frac{4}{K}}$, we get

$$\sum_{k=1}^{K} \left\langle c_{\mathrm{true}} - c^k, \mathbf{e}_{s_{L^k}^k} - \widehat{d^{\pi_\mathrm{E}}} \right\rangle \leq 2\sqrt{K}.$$

Then, we can handle the remaining two terms. In particular $\sum_{k=1}^{K} \left\langle c_{\mathrm{true}} - c^k, d^{\pi^k} - \mathbf{e}_{s_{L^k}^k} \right\rangle$ is the sum of a martingale difference sequence. Therefore, applying the Azuma-Hoeffding inequality it holds that with probability $1 - \delta$

$$\sum_{k=1}^{K} \left\langle c_{\mathrm{true}} - c^k, d^{\pi^k} - \mathbf{e}_{s_{L^k}^k} \right\rangle \leq \sqrt{2K \log(1/\delta)}$$

where we used that $\left| \left\langle c_{\mathrm{true}} - c^k, d^{\pi^k} - \mathbf{e}_{s_{L^k}^k} \right\rangle \right| \leq 2$ for all $k \in [K]$. Finally for the expert concentration term, we have that

$$\sum_{k=1}^{K} \left\langle c_{\mathrm{true}} - c^k, \widehat{d^{\pi_\mathrm{E}}} - d^{\pi_\mathrm{E}} \right\rangle \leq K \sqrt{|\mathcal{S}| \, |\mathcal{A}|} \left\| d^{\pi_\mathrm{E}} - d^{\pi_\mathrm{E}} \right\|_\infty$$

Then, for any fixed state action pair $s, a$ with probability $1 - \delta/(|\mathcal{S}| \, |\mathcal{A}|)$ by Azuma-Hoeffding inequality it holds that

$$d^{\pi_\mathrm{E}}(s) - d^{\pi_\mathrm{E}}(s) = \frac{1}{|\mathcal{D}_{\pi_\mathrm{E}}|} \sum_{s' \in \mathcal{D}_{\pi_\mathrm{E}}} \mathbb{1}_{\{s'=s\}} - d^{\pi_\mathrm{E}}(s) \leq \sqrt{\frac{\log(|\mathcal{S}| \, |\mathcal{A}| \, \delta^{-1})}{2 \, |\mathcal{D}_{\pi_\mathrm{E}}|}}$$

Therefore, by a union bound it holds that with probability $1 - \delta$,

$$\left\| d^{\pi_\mathrm{E}} - d^{\pi_\mathrm{E}} \right\|_\infty \leq \sqrt{\frac{\log(|\mathcal{S}| \, |\mathcal{A}| \, \delta^{-1})}{2 \, |\mathcal{D}_{\pi_\mathrm{E}}|}}.$$

Putting together, the bounds on the three terms allow to conclude the proof. $\qquad \square$

## C. Technical Lemmas

**Lemma C.1.** *Consider the MDP $M = (\mathcal{S}, \mathcal{A}, P, c, \boldsymbol{\nu}_0, \gamma)$ and two policies $\pi, \pi' : \mathcal{S} \to \Delta_\mathcal{A}$. Then consider for any $\widehat{Q} \in \mathbb{R}^{|\mathcal{S}||\mathcal{A}|}$ and $\widehat{V}^\pi(s) = \left\langle \pi(\cdot|s), \widehat{Q}(s, \cdot) \right\rangle$ and $Q^{\pi'}, V^{\pi'}$ be respectively the state-action and state value function of the policy $\pi$ in MDP $M$. Then, it holds that $(1 - \gamma) \left\langle \boldsymbol{\nu}_0, \widehat{V}^\pi - V^{\pi'} \right\rangle$ equals*

$$\left\langle d^{\pi'}, \widehat{Q} - c - \gamma P \widehat{V}^\pi \right\rangle + \mathbb{E}_{s \sim d^{\pi'}} \left[ \left\langle \widehat{Q}(s, \cdot), \pi(\cdot|s) - \pi'(\cdot|s) \right\rangle \right].$$

*Proof.*

$$\left\langle d^{\pi'}, \hat{Q} \right\rangle = \left\langle d^{\pi'}, \hat{Q} - c - \gamma P \hat{V}^\pi \right\rangle + \left\langle d^{\pi'}, c + \gamma P \hat{V}^\pi \right\rangle$$

Then, using the property of occupancy measure we have that $\left\langle d^{\pi'}, c \right\rangle = (1 - \gamma) \left\langle \boldsymbol{\nu}_0, V^{\pi'} \right\rangle$ where $V^{\pi'}$ is the value function of the policy $\pi'$ in the MDP. Then, it holds that

$$\left\langle d^{\pi'}, \hat{Q} \right\rangle = \left\langle d^{\pi'}, \hat{Q} - c - \gamma P \hat{V}^\pi \right\rangle + (1 - \gamma) \left\langle \boldsymbol{\nu}_0, V^{\pi'} \right\rangle + \left\langle d^{\pi'}, \gamma P \hat{V}^\pi \right\rangle$$

$$= \left\langle d^{\pi'}, \hat{Q} - c - \gamma P \hat{V}^\pi \right\rangle + (1-\gamma) \left\langle \boldsymbol{\nu}_0, V^{\pi'} \right\rangle + \left\langle \gamma P^T d^{\pi'}, \hat{V}^\pi \right\rangle$$

$$= \left\langle d^{\pi'}, \hat{Q} - c - \gamma P \hat{V}^\pi \right\rangle + (1-\gamma) \left\langle \boldsymbol{\nu}_0, V^{\pi'} \right\rangle + \left\langle E^T d^{\pi'} - (1-\gamma)\boldsymbol{\nu}_0, \hat{V}^\pi \right\rangle$$

$$= \left\langle d^{\pi'}, \hat{Q} - c - \gamma P \hat{V}^\pi \right\rangle + (1-\gamma) \left\langle \boldsymbol{\nu}_0, V^{\pi'} - \hat{V}^\pi \right\rangle + \left\langle E^T d^{\pi'}, \hat{V}^\pi \right\rangle.$$

Rearranging and using the definition of $\widehat{V}^\pi$ yields the conclusion. $\qquad\square$

**Lemma C.2.** *Let us assume that $L^k \le L_{\max}$ for all $k \in [K]$. Then, it holds that with probability $1 - \delta$*

$$\sum_{k=1}^K \mathbb{E}_{s,a \sim d^{\pi^k}} \left[ \frac{1}{N^k(s,a)/L + 1} \right] \le 2L |\mathcal{S}| |\mathcal{A}| \log(KL_{\max}) + 4 \log(2KL_{\max}/\delta)$$

*Proof.* Let us assume that $L^k \le L_{\max}$ for all $k \in [K]$. Invoking (**?**)Lemma D.4]cohen2019learning, it holds that with probability $1 - \delta$

$$\sum_{k=1}^K \mathbb{E}_{s,a \sim d^{\pi^k}} \left[ \frac{1}{N^k(s,a)/L + 1} \right] \le 2 \sum_{k=1}^K \sum_{t=1}^{L^k} \frac{1}{N^k(s_t^k, a_t^k)/L + 1} + 4 \log(2KL_{\max}/\delta)$$

$$\le 2 \sum_{s,a \in \mathcal{S} \times \mathcal{A}} \sum_{k=1}^K \sum_{t=1}^{L^k} \frac{\mathbb{1}_{\{s,a=s_t^k,a_t^k\}}}{N^k(s,a)/L + 1} + 4 \log(2KL_{\max}/\delta)$$

$$\le 2L \sum_{s,a \in \mathcal{S} \times \mathcal{A}} \sum_{k=1}^K \sum_{t=1}^{L^k} \frac{\mathbb{1}_{\{s,a=s_t^k,a_t^k\}}}{N^k(s,a) + 1} + 4 \log(2KL_{\max}/\delta)$$

$$\le 2L \sum_{s,a \in \mathcal{S} \times \mathcal{A}} \sum_{k=1}^K \sum_{t=1}^{L^k} \frac{\mathbb{1}_{\{s,a=s_t^k,a_t^k\}}}{\sum_{\tau=1}^k \sum_{t=1}^{L_\tau} \mathbb{1}_{\{s,a=s_t^\tau,a_t^\tau\}} + 1} + 4 \log(2KL_{\max}/\delta)$$

$$\le 2L \sum_{s,a \in \mathcal{S} \times \mathcal{A}} \log \left( \sum_{k=1}^K \sum_{t=1}^{L^k} \mathbb{1}_{\{s,a=s_t^k,a_t^k\}} \right) + 4 \log(2KL_{\max}/\delta)$$

$$\le 2L |\mathcal{S}| |\mathcal{A}| \log(KL_{\max}) + 4 \log(2KL_{\max}/\delta)$$

where we used Lemma C.3 for $f(x) = x^{-1}$. $\qquad\square$

**Lemma C.3.** *Let $a_0 \ge 0$ and $f[0,\infty) \to [0,\infty)$ be a non increasing function , then*

$$\sum_{t=1}^T \alpha_t f\left(a_0 + \sum_{t=1}^T \alpha_t\right) \le \int_{a_0}^{\sum_{t=1}^T a_t} f(x)dx$$

*Proof.* See (Orabona, 2023) Lemma 4.13. $\qquad\square$

**Lemma C.4.** *The sequence of policies $\left\{\pi^k\right\}_{k=1}^K$ generated by Algorithm 2 and let $d^\pi$ denote the occupancy measure for the policy $\pi$. Then it holds that*

$$\forall k \in [K] \quad \left\| d^{\pi^k} - d^{\pi^{k+1}} \right\|_1 \le \frac{\eta Q_{\max}}{(1-\gamma)}$$

*Proof.* By Lemma A.1 in (Sun et al., 2019a) it holds that

$$\left\| d^{\pi^k} - d^{\pi^{k+1}} \right\|_1 \le \frac{1}{1-\gamma} \mathbb{E}_{x \sim d^{\pi^k}} \left[ \left\| \pi^k(\cdot|x) - \pi^{k+1}(\cdot|x) \right\|_1 \right]$$

Then, we notice that by 1-strong convexity of the KL divergence it holds that

$$\frac{1}{2}\mathbb{E}_{x\sim d^{\pi^k}}\left[\left\|\pi^k(\cdot|x)-\pi^{k+1}(\cdot|x)\right\|_1^2\right]\leq\frac{1}{2}\mathbb{E}_{x\sim d^{\pi^k}}\left[D_{KL}(\pi^{k+1}(\cdot|x),\pi^k(\cdot|x))\right]$$

$$\leq\frac{1}{2}\mathbb{E}_{x\sim d^{\pi^k}}\sum_{a\in\mathcal{A}}\pi^{k+1}(a|x)\left(-\eta Q_k(x,a)-\log\left(\sum_{a\in\mathcal{A}}\pi^k(a|x)\exp(-\eta Q_k(x,a))\right)\right)$$

$$=-\frac{\eta}{2}\mathbb{E}_{x\sim d^{\pi^k}}\sum_{a\in\mathcal{A}}\pi^{k+1}(a|x)Q_k(x,a)-\frac{1}{2}\mathbb{E}_{x\sim d^{\pi^k}}\log\left(\sum_{a\in\mathcal{A}}\pi^k(a|x)\exp(-\eta Q_k(x,a))\right)$$

$$\leq-\frac{\eta}{2}\mathbb{E}_{x\sim d^{\pi^k}}\sum_{a\in\mathcal{A}}\pi^{k+1}(a|x)Q_k(x,a)+\frac{\eta}{2}\mathbb{E}_{x\sim d^{\pi^k}}\sum_{a\in\mathcal{A}}\pi^k(a|x)Q_k(x,a)$$

where the last inequality follows by Jensen's inequality and convexity of $-\log$. Hence, we continue the upper bound as follows

$$\frac{1}{2}\mathbb{E}_{x\sim d^{\pi^k}}\left[\left\|\pi^k(\cdot|x)-\pi^{k+1}(\cdot|x)\right\|_1^2\right]=\frac{\eta}{2}\mathbb{E}_{x\sim d^{\pi^k}}\sum_{a\in\mathcal{A}}Q_k(x,a)\cdot(\pi^k(\cdot|x)-\pi^{k+1}(\cdot|x))$$

$$\leq\frac{\eta Q_{\max}}{2}\cdot\mathbb{E}_{x\sim d^{\pi^k}}\left[\left\|\pi^k(\cdot|x)-\pi^{k+1}(\cdot|x)\right\|_1\right]$$

Which implies, by Jensen's inequality and diving both sides by $\frac{1}{2}\mathbb{E}_{x\sim d^{\pi^k}}\left[\left\|\pi^k(\cdot|x)-\pi^{k+1}(\cdot|x)\right\|_1\right]$ that

$$\mathbb{E}_{x\sim d^{\pi^k}}\left[\left\|\pi^k(\cdot|x)-\pi^{k+1}(\cdot|x)\right\|_1\right]\leq\eta Q_{\max}.$$

$\square$

**Lemma C.5.** *Samuelson's inequality* *Let us consider $L$ scalars $\{X_\ell\}_{\ell=1}^L$ and denote the sample mean as $\bar{X}=L^{-1}\sum_{\ell=1}^L X_\ell$ and the empirical standard deviation as $\hat{\sigma}=\sqrt{\frac{\sum_{\ell=1}^L(X_\ell-\bar{X})^2}{L-1}}$, then it holds that*

$$\bar{X}-\sqrt{L-1}\hat{\sigma}\leq X_\ell\leq\bar{X}+\sqrt{L-1}\hat{\sigma}\quad\forall\ \ell\in[L]$$

*Proof.* Let us consider an arbitrary vector $v\in\mathbb{R}^L$. Then, we have that $\|v\|_\infty\leq\|v\|_2$. At this point let us consider $v=[X_1-\bar{X},\ldots,X_L-\bar{X}]^T$. Moreover, let us define as $\ell^\star$ the index such that $\|v\|_\infty=\left|X_{\ell^\star}-\bar{X}\right|$. Then, we have that for all $\ell\in[L]$,

$$\left|X_\ell-\bar{X}\right|\leq\left|X_{\ell^\star}-\bar{X}\right|\leq\sqrt{\sum_{\ell=1}^L(X_\ell-\bar{X})^2}=\sqrt{L-1}\hat{\sigma}.$$

Therefore, rewriting the absolute value it holds that

$$\bar{X}-\sqrt{L-1}\hat{\sigma}\leq X_\ell\leq\bar{X}+\sqrt{L-1}\hat{\sigma}$$

$\square$

The next lemma says that the effective horizon in the original MDP and the binarized MDP is equal up to a $\log_2(|\mathcal{S}|)$ factor.

**Lemma C.6.** *It holds that*

$$\frac{1}{1-\gamma^{1/\log_2|\mathcal{S}|}}\leq\frac{\log_2|\mathcal{S}|+2}{1-\gamma}.$$

*Proof.*

$$\frac{1}{1-\gamma^{1/\log_2|\mathcal{S}|}}=\frac{1}{1-\gamma}\frac{1-\gamma}{1-\gamma^{1/\log_2|\mathcal{S}|}}$$

$$= \frac{1}{1-\gamma} \frac{1 - \gamma_{\text{bin}}^{\log_2 |\mathcal{S}|}}{1 - \gamma_{\text{bin}}}$$

$$= \frac{1}{1-\gamma} \sum_{t=0}^{\log_2 |\mathcal{S}|+1} \gamma_{\text{bin}}^t$$

$$\leq \frac{\log_2 |\mathcal{S}| + 2}{1-\gamma}.$$

$\square$

## D. Implementation details

**Environment:** We use the Hopper-v5, Ant-v5, HalfCheetah-v5, and Walker2d-v5 environments from OpenAI Gym.

**Expert Samples:** The expert policy is trained using SAC. The training configuration uses 3000 epochs. The agent explores randomly for the first 10 episodes before starting policy learning. A replay buffer of 1 million experiences is used, with a batch size of 100 and a learning rate of 1e-3. The temperature parameter ($\alpha$) is set to 0.2. The policy updates occur every 50 steps, with 1 update per interval. After training 64 experts trajectories are collected to be used later for the agent training. The cumulative reward of the expert policies in the different environments are as given in the following table.

Table 1: Expert returns

| Method | Ant-v5 | Hopper-v5 | Humanoid-v5 | Walker2d-v5 |
|---|---|---|---|---|
| Expert return | 4061.41 ± 730.58 | 3500.87 ± 4.33 | 5237.48 ± 414.69 | 5580.39 ± 20.30 |

**IL algorithms implementation:** Our starting code base is taken from the repository of `f-IRL`[7] (Ni et al., 2021), and the implementation of the other algorithms are based on this one. For more details about the implementation please refer to our repository. The most important hyperparameters are reported in Table 2

- **ML-IRL, f-IRL and rkl:** These algorithms were already implemented in the `f-IRL` repository. The method leverages SAC as the underlying reinforcement learning algorithm and different type of objectives for the cost update. The multi-Q-network exploration bonus is implemented inside the SAC update, there we keep track of multiple Q-networks and use their mean and standard deviation to update the policy. The clipping is applied on the standard deviation which serves as the exploration bonus.

- **CSIL:** We started from the `f-IRL` implementation, maintaining the same hyperparameters for a fair comparison. The key modification was removing reward model training from the RL loop, instead training it only once before entering the loop using behavioral cloning and $L_2$ normalization, after which the reward model remained fixed throughout the training.

- **OPT-AIL (state-only and state-action):** We started from the implementation of `ML-IRL` and added the OPT-AIL exploration bonus, incorporating optimism-regularized Bellman error minimization for Q-value functions as described in the original article (Xu et al., 2024).

The updated Q-loss can be formulated as:

$$\mathcal{L}_Q = \mathbb{E}\left[ \left( Q_\theta(s,a) - (r + \gamma(1-d)(Q_{\bar{\theta}}(s',a') - \alpha \log \pi(a'|s'))) \right)^2 \right] - \lambda \mathbb{E}[Q_\theta(s,a)] \tag{3}$$

Where: - $Q_\theta$ is the current Q-network - $Q_{\bar{\theta}}$ is the target Q-network - $r$ is the learned reward - $\gamma$ is the discount factor - $d$ is the done flag - $\alpha$ is the entropy coefficient - $\lambda$ is the optimism regularization parameter - The expectation is taken across the data distribution $\mathcal{D}$ sampled from the replay buffer, which includes state-action-reward-next state-done transitions and individual state-action pairs.

---

[7]https://github.com/twni2016/f-IRL/tree/main

Table 2: Core Hyperparameters Across Environments

| Parameter | Walker2d | Humanoid | Hopper | Ant |
|---|---|---|---|---|
| Number of Iterations | 1.5 M | 1 M | 1 M | 1.2 M |
| Reward Network size | [64, 64] | [64, 64] | [64, 64] | [128, 128] |
| Policy Network size | [256, 256] | [256, 256] | [256, 256] | [256, 256] |
| Reward Learning Rate | 1e-4 | 1e-4 | 1e-4 | 1e-4 |
| SAC Learning Rate | 1e-3 | 1e-3 | 1e-3 | 1e-3 |

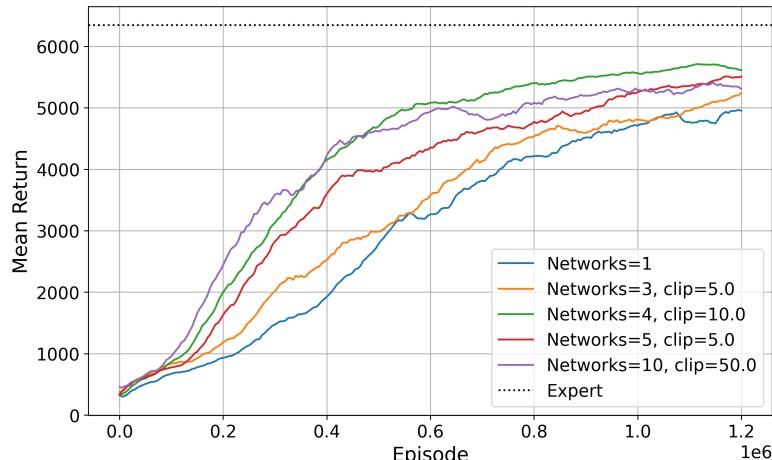

Figure 5: Mean return of ML-IRL Ant-v5 with a different number of neural networks. The grid search for the clipping values was performed over the following values [0.1 0.5 1 5 10 50]. Results are averaged over 3 seeds.

- **SQIL** (Reddy et al., 2019b): was implemented by initializing a replay buffer with expert trajectories and assigning them a reward of 1, while collecting additional on-policy trajectories from the agent's current policy with a reward of 0. During training, the `SAC` agent learns from both expert and agent-generated transitions, effectively learning to imitate expert behavior through the asymmetric reward structure. The agent updates its policy by sampling from this mixed replay buffer, where the expert transitions provide a high-reward signal to guide the learning process.

- **GAILs:** we used the implementation available from `Stable-Baselines3` (Raffin et al., 2021). This is the only method not based on `SAC`.

### D.1. Hyperparameters tuning

To determine how different numbers of neural networks changed the performance, we conducted an ablation study on both the clipping value and the number of neural networks. We performed this analysis for the ML-IRL algorithm on the Ant-V5 environment. We chose this environment since previous experiments showed that its higher complexity led to higher variance in the performance of different algorithms. It was necessary to perform a grid search on the number of neural networks because we noticed that different numbers of neural networks preferred different clipping values.

The results are reported in Figure 5. As we observed, in all cases, adding more neural networks leads to better performances. However, this improvement does not increase proportionally with the number of neural networks; in fact, the run with 10 neural networks is outperformed by the one with 4. This led us to select 4 as the fixed value of neural networks, also justified by the much slower training time of the 10-network case.

For every environment and algorithm, we performed a grid search over different clipping values. The range of clipping values varied across algorithms. Figure 6 shows the different values used in the search and their impact on performance. The difference in performance across clipping values is small in simpler environments (e.g., Hopper or Walker2d) while it

becomes more evident in more complex environments with larger state-action spaces. These plots also show the necessity of the clipping for the exploration bonus. In most environments and algorithms, when a large clipping value is applied, it leads to performance degradation.

Empirically, the Q-network's standard deviation diverges due to unclipped Q-values. Without value clipping, high Q-values for specific state-action pairs increase the probability of being visited, causing more of these pairs to accumulate in the replay buffer across different rollouts and potentially amplifying the standard deviation across the q-network for the next update. This justifies the necessity of a clipping value on the exploration bonus.

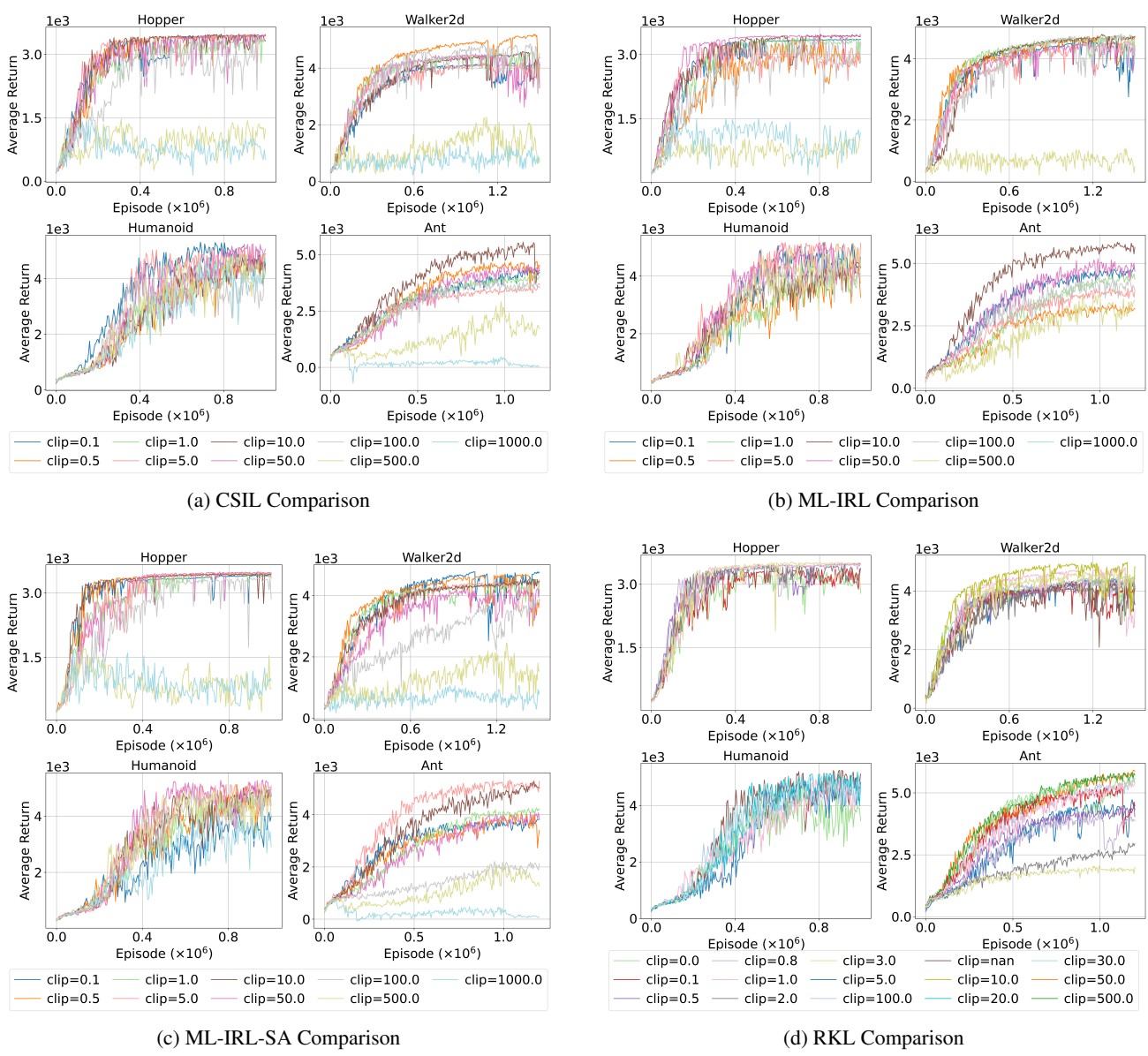

Figure 6: Comparison of clipping values across different environments, showing the effect on average return for each environment.

# E. Experiments with single expert trajectory

Here we report the we report the result of the experiments using a single trajectory. Our findings indicate that the performance remained consistent regardless of the number of trajectories used and the performance are comparable to the ones with 16 trajectories. Notable differences in performance improvement were observed in Humanoid-v5 and ant state environments in the state only settings, where a more pronounced gap was evident.

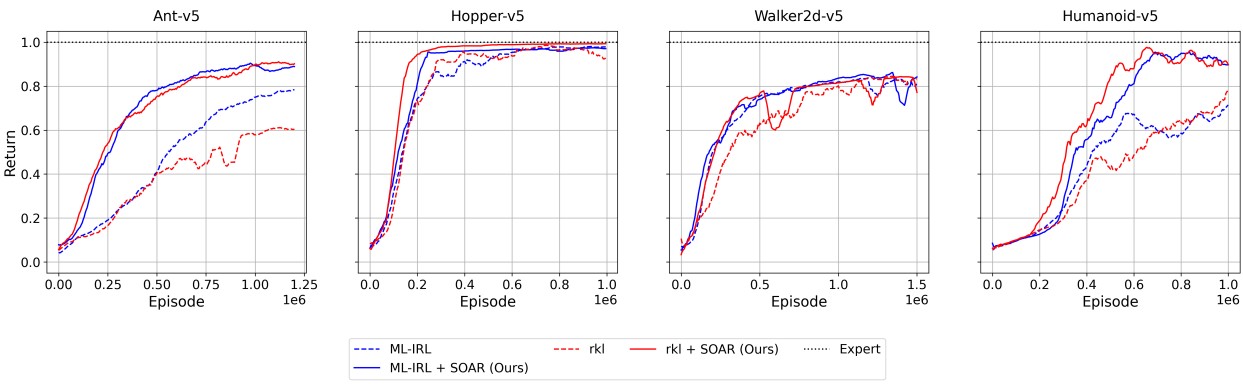

Figure 7: **Experiments from State-Only Expert Trajectories**. 1 expert trajectories, average over 3 seeds, $L = 4$ Clipping values $\sigma$ - ML-IRL: [Ant: 0.1, Hopper: 0.1, Walker2d: 50.0, Humanoid: 0.5], rkl: [Ant: 0.1, Hopper: 0.5, Walker2d: 1.0, Humanoid: 50.0]

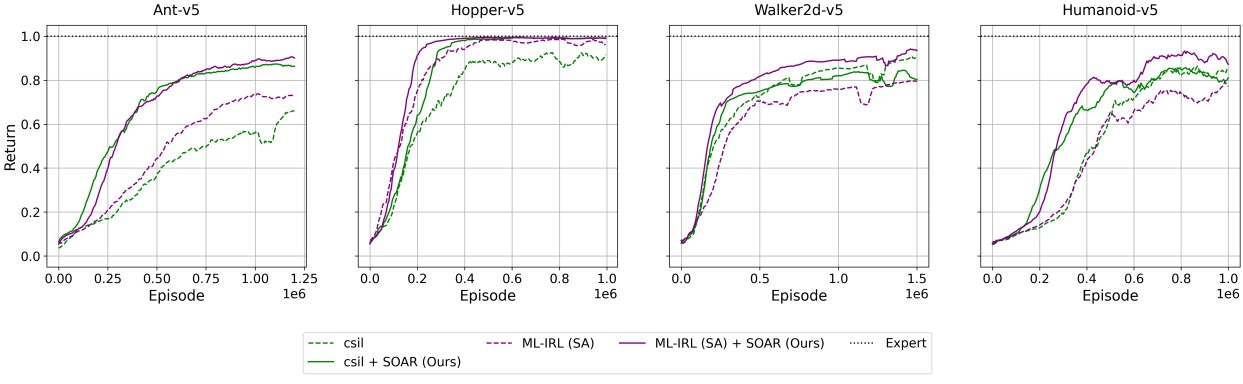

Figure 8: **Experiments from State-Action Expert Trajectories**. 1 expert trajectories, average over 3 seeds, $L = 4$. Clipping values $\sigma$ - CSIL: [Ant: 0.5, Hopper: 50.0, Walker2d: 0.1, Humanoid: 0.1], ML-IRL(SA): [Ant: 0.1, Hopper: 0.5, Walker2d: 0.1, Humanoid: 1.0]

# F. Omitted Pseudocodes

This section introduces the omitted pseudocodes to clarify the implementation of the algorithms based on SOAR. We first give a pseudocode (see Algorithm 6) that mirrors Algorithm 1 in the setting where deep neural network approximation is needed due to the continuous structure of the state-action space. The critic training is the same as in the standard SAC (Haarnoja et al., 2018) but we report it in Algorithm 7 for safe completeness. Notice that we adopt the double critic training originally proposed in (van Hasselt et al., 2015) to avoid an excessive underestimation of the critics value[8].

---

[8]Notice that (van Hasselt et al., 2015) talks about excessive overestimation of the prediction target in the critic training rather than underestimation. This difference is due to the fact that their paper casts RL as reward maximization while we adopt a cost minimization perspective. For the same reason we take the maximum between the two critics rather than the minimum as done in (van Hasselt et al., 2015).

---

**Algorithm 6** Base Method + SOAR pseudocode

---

**Require:** Policy step size $\eta$, cost step size $\alpha$, expert dataset $\mathcal{D}_{\tau_E}$, discount factor $\gamma$, maximum standard deviation parameter $\sigma$,

1: Initialize actor network $\pi_{\psi^1}$ randomly.
2: Initialize the cost network $c_{w^1}$ randomly.
3: Initialize the $L$ critics $\{Q_{\theta_1^1}, \ldots, Q_{\theta_L^1}\}$ randomly.
4: Initialize the $L$ target critics $\{Q_{\theta_1^{1,\text{targ}}}, \ldots, Q_{\theta_L^{1,\text{targ}}}\}$ randomly.
5: Initialize $L$ empty replay buffers $\{\mathcal{D}_\ell^k\}_{\ell=1}^L$. (One for each critic)
6: **for** $k = 1$ to $K$ **do**
7:    $\tau_\ell^k \leftarrow \text{COLLECTTRAJECTORY}(\pi)$ for each $\ell \in [L]$.
8:    Add $\tau_\ell^k$ to replay buffer $\mathcal{D}_\ell^k \leftarrow \mathcal{D}_\ell^{k-1} \cup \tau_\ell^k$.
9:    Let $\mathcal{D}^k = \cup_{\ell=1}^L \mathcal{D}_\ell^k$
10:    $c_{w^k} \leftarrow \text{UPDATECOST}(c_{w^{k-1}}, \mathcal{D}_{\pi_E}, \mathcal{D}^k, \alpha)$ using the Base Method (such as CSIL, $f$-IRL or ML-IRL ).
11:    **for** $\ell = 1$ to $L$ **do**
12:      $Q_{\theta_\ell^{k+1}}, Q_{\theta_\ell^{k+1,\text{targ}}} = \text{UPDATECRITICS}(\mathcal{D}_\ell^k, \pi_{\psi^k}, \eta, \gamma, c_{\theta^k})$
13:    **end for**
14:    $\{Q^{k+1}(s,a)\}_{s,a \in \mathcal{D}^k} = \text{OPTIMISTICQ-NN}(\mathcal{D}^k, \{Q_{\theta_\ell^{k+1}}\}_{\ell=1}^L, \sigma)$    (see Algorithm 5)
15:    Define the loss $\mathcal{L}_\pi^k = \frac{1}{|\mathcal{D}^k|} \sum_{s,a \in \mathcal{D}^k} \left(-\eta \log \pi_{\psi^k}(a|s) + Q^{k+1}(s,a)\right)$.
16:    Update policy weights to $\psi^{k+1}$ using Adam (Kingma & Ba, 2015) on the loss $\mathcal{L}_\pi^k$.
17: **end for**
18: **Return** $\pi$

---

**Algorithm 7** UPDATECRITICS

---

**Require:** $\mathcal{D}_\ell^k, \pi_{\psi^k}, \alpha, \gamma, c_w$

1: Let $\mathcal{B} = \{s_i, a_i, r, s_i', \text{done}_i\}_{i=1}^N$ be a minibatch sampled from $\mathcal{D}$
2: $a_i' \leftarrow \pi(s_i')$ for all $i \in [N]$.
3: Define $Q_{\theta_\ell^{k,\text{targ}}}(s_i, a_i) \leftarrow \max\left(Q_{\theta_\ell^{k,\text{targ},(1)}}(s_i, a_i), Q_{\theta_\ell^{k,\text{targ},(2)}}(s_i, a_i)\right)$ for all $s_i, a_i \in \mathcal{B}$.
4: $\text{Backup}_i \leftarrow c_w(s_i, a_i) + \gamma(1 - \text{done}_i)\left(Q_{\theta_\ell^{k,\text{targ}}}(s_i, a_i) + \alpha \log \pi_{\psi^k}(a_i'|s_i')\right)$
5: $\mathcal{L}_{\theta_\ell^{k,(1)}} = \frac{1}{N} \sum_{i=1}^N \left(Q_{\theta_\ell^{k,(1)}}(s_i, a_i) - \text{Backup}_i\right)^2$
6: $\mathcal{L}_{\theta_\ell^{k,(2)}} = \frac{1}{N} \sum_{i=1}^N \left(Q_{\theta_\ell^{k,(2)}}(s_i, a_i) - \text{Backup}_i\right)^2$
7: $\theta_\ell^{k+1,(1)} \leftarrow \theta_\ell^{k,(1)} - \eta_Q \nabla \mathcal{L}_{\theta_\ell^{k,(1)}}$
8: $\theta_\ell^{k+1,(2)} \leftarrow \theta_\ell^{k,(2)} - \eta_Q \nabla \mathcal{L}_{\theta_\ell^{k,(2)}}$.
9: $\theta^{k+1,\text{targ},(1)} \leftarrow (1 - \tau_{\text{targ}})\theta^{k,\text{targ},(1)} + \tau_{\text{targ}}\theta^{k,(1)}$.
10: $\theta^{k+1,\text{targ},(2)} \leftarrow (1 - \tau_{\text{targ}})\theta^{k,\text{targ},(2)} + \tau_{\text{targ}}\theta^{k,(2)}$.
11: $Q_{\theta_\ell^{k+1}}(s,a) = \max\left(Q_{\theta_\ell^{k+1,(1)}}(s,a), Q_{\theta_\ell^{k+1,(2)}}(s,a)\right)$ for all $s, a \in \mathcal{D}^k$.
12: $Q_{\theta_\ell^{k+1,\text{targ}}}(s,a) = \max\left(Q_{\theta_\ell^{k+1,\text{targ},(1)}}(s,a), Q_{\theta_\ell^{k+1,\text{targ},(2)}}(s,a)\right)$ for all $s, a \in \mathcal{D}^k$.
13: **return** $Q_{\theta_\ell^{k+1}}, Q_{\theta_\ell^{k+1,\text{targ}}}$.

---

### F.1. Instantiating the cost update

We show after how the algorithmic template in Algorithm 1 captures different imitation learning algorithms just changing the cost update. For example, $f$-IRL with the reversed KL divergence (RKL) can be seen as Algorithm 1 with the cost update described in Algorithm 8. Moreover, our SOAR+RKL is obtained plugging in the cost update in Algorithm 8 in Algorithm 6.

---

**Algorithm 8** UPDATECOST for RKL ($f$-IRL for reversed KL divergence) (Ni et al., 2021)

---

**Require:** $c, \mathcal{D}_{\pi_E}, \mathcal{D}_{\pi^k}, \alpha$, divergence generating function $f(x) = -\log(x)$ for the reversed KL divergence, prior distribution over trajectories $p(\tau)$.

1: $\rho_w(\tau) = \frac{1}{Z} p(\tau) e^{-c_w(\tau)}$
2: $\chi^\star \leftarrow \arg\max_\omega \mathbb{E}_{s \sim \mathcal{D}_{\pi_E}}[\log D_\chi(s)] + \mathbb{E}_{s \sim \mathcal{D}^k}[\log(1 - D_\chi(s))]$
3: Estimate the density ratio:
4: $\frac{\rho_E(s)}{\rho_w(s)} = \frac{D_{\chi^\star}(s)}{1 - D_{\chi^\star}(s)}$
5: Compute the stochastic gradient

$$
\widehat{\nabla_w} = \frac{1}{T} \mathbb{E}_{\tau \sim \rho_w} \left[ \sum_{t=1}^{T} h_f \left( \frac{\rho_E(s_t)}{\rho_w(s_t)} \right) \cdot \left( -\sum_{t=1}^{T} \nabla_\theta c_w(s_t) \right) \right]
$$
$$
- \frac{1}{T} \mathbb{E}_{\tau \sim \rho_w} \left[ \sum_{t=1}^{T} h_f \left( \frac{\rho_E(s_t)}{\rho_\theta(s_t)} \right) \right] \cdot \mathbb{E}_{\tau \sim \rho_\theta} \left[ \left( -\sum_{t=1}^{T} \nabla_\theta c_w(s_t) \right) \right]
$$

6: $w \leftarrow w - \alpha \widehat{\nabla_w}$
7: **Return** $c_w$

---

Next, we present the cost update for the algorithm ML-IRL (Zeng et al., 2022). We present it for the state-action version. The state-only version is obtained simply omitting the action dependence everywhere.

---

**Algorithm 9** UPDATECOST for ML-IRL (State-Action version) (Zeng et al., 2022)

---

**Require:** $c_w, \mathcal{D}_{\pi_E}, \tau^k = \left\{ s_t^k, a_t^k \right\}_{t=1}^{L^k}, \alpha$.

1: Sample a state-action trajectory $\tau_E = \left\{ s_t^E, a_t^E \right\}_{t=1}^{L_E}$ from the expert dataset $\mathcal{D}_{\pi_E}$ where $L_E$ is a geometric random variable with parameter $(1 - \gamma)^{-1}$.
2: Compute the stochastic loss

$$
\widehat{\mathcal{L}}_w = \sum_{t=0}^{L_E} \gamma^t c_w(s_t^E, a_t^E) - \sum_{t=0}^{L^k} \gamma^t c_w(s_t^k, a_t^k)
$$

3: $w \leftarrow w - \alpha \nabla_w \widehat{\mathcal{L}}_w$
4: **Return** $c_w$

---

To conclude, we present the cost update for CSIL. Notice that since the cost used by CSIL does not leverage the information of the policy at iteration $k$ we can move the cost update before the main loop and keep a constant cost function fixed during the training of the policy. Notice that the CSIL the reward is simply given by the log probabilities learned by the behavioural cloning policy. Therefore the reward parameters in CSIL coincides with the parameters of the behavioral cloning policy network. We point out that using a reward of this form is similarly done in (Vieillard et al., 2020). We plan to explore further the connection between Munchausen RL and CSIL in future work.

---

**Algorithm 10** UPDATECOST for CSIL (Watson et al., 2023)

---

**Require:** $\mathcal{D}_{\pi_E}$.

1: Compute the behavioral cloning policy finding an approximate solution to the following problem.

$$
w^\star = \arg\max_w \sum_{s,a \in \mathcal{D}_{\pi_E}} \log \pi_w(a|s)
$$

2: **Return** $c_{w^\star}(s, a) = -\log \pi_{w^\star}(a|s)$

---

# G. Details for the experiment on a hard exploration task

The environment used for this experiment coincides with the construction used in the lower bound for the number of environment interaction in (Moulin et al., 2025, Theorem 19). This is a simple two states MDP ( a low reward state and a high reward state ) with 20 actions per state. From the high reward state all actions are identical. From the low reward state, all actions are identical but the one chosen by the deterministic expert which has just a slightly higher probability to lead to the high reward state from the low reward state. Even observing the expert state occupancy measure perfectly, it is difficult for the learner to find out which is the action which the expert took. That is because all actions are almost identical but one.

