# OpenReview forum: "IL-SOAR : Imitation Learning with Soft Optimistic  Actor cRitic"
_ICML.cc/2025/Conference — ICML 2025 poster_

### Official Review · Reviewer_e44U · 2025-03-11

**Overall Recommendation:** 4

**Summary:**

The paper presents an imitation learning method for the tabular setting and establishes a probabilistic bound on its regret, considering both, regret due to the policy updates and the cost updates. The method is based on the principle of optimism in the face of uncertainty by learning a transition matrix the transition probabilities - thereby reducing the expected cost. Two different options, "Min" and "Mean-Std",  are presented for obtaining optimistic Q values based on this transition matrix. The algorithm returns a mixture policy, with regret below $\epsilon$ with high probability, requiring $\mathcal{O}(\epsilon^{-2})$ iterations/environment interactions, and $\mathcal{O}(\epsilon^{-2})$ expert interactions.

The paper also proposes an extension for imitation learning methods based on SAC that uses ensembles to obtain optimistic Q-functions. The experiments show, that using these optimistic Q-functions can clearly improve the performance of the tested methods (Coherent Soft Imitation Learning, Maximum Likelihood imitation learning, RKL).

## update after rebuttal

Based on the rebuttal and the other reviews, I do not see the need for any updates.

**Claims And Evidence:**

The main claims, e.g. the regret bound (Theroem 3.4) are supported by the provided derivations.

**Essential References Not Discussed:**

Very recently, Moulin, Neu and Viano have submitted the related work "Optimistically Optimistic Exploration for Provably Efficient Infinite-Horizon Reinforcement and Imitation Learning" to arxiv, which could be already discussed in the manuscript.

**Experimental Designs Or Analyses:**

The experimental design is sound, however, it uses too few seeds (5). It would also be useful to compare the performance for different data set sizes (the appendix does contain experiments with a single expert trajectory, though).

**Methods And Evaluation Criteria:**

The method of using optimism to improve exploration and thereby sample efficiency makes sense. The mujoco locomotion experiments are suitable as they are commonly used by related methods, allowing for a fair comparison.

**Other Comments Or Suggestions:**

In Algorithm 1, line 11, it seems like $K$ should rather be $k$.

**Other Strengths And Weaknesses:**

The main contribution seems to be the theoretical analysis for the tabular algorithm. The empirical results in the continuous setting, IL-SOAR, are of practical interest, but this algorithm is quite different from the tabular one. Hence, the different contributions are almost orthogonal (although optimistic Q-functions is, of course, a common theme).

**Questions For Authors:**

Line 372 refers to an "attainable range". How is this defined?

**Relation To Broader Scientific Literature:**

Exploration strategies, also based on optimism, are very popular in reinforcement learning, but are much less common in imitation learning, presumably due to the fact that the demonstrations may guide the agent. However, the submission shows, that optimism in the face of uncertainty can result in significant improvements in sample efficiency.

**Theoretical Claims:**

I did not check the derivations in the appendix, but I could not find any issues with the theoretical claims in the main manuscript.

---

> ### Author Rebuttal · Authors · 2025-03-27
>
> Dear reviewer,
>
> Thanks a lot for appreciating our work !
>
> Please find our responses to your comments below.
>
> ***The experimental design is sound, however, it uses too few seeds (5). It would also be useful to compare the performance for different data set sizes (the appendix does contain experiments with a single expert trajectory, though).***
>
> We will increase the number of seeds. As can be appreciated in the appendix it seems that even with just 1 expert trajectory SOAR improves against the non exploratory counterparts.
>
>
> ***Exploration strategies, also based on optimism, ... in sample efficiency.***
>
> We definitely agree with the reviewer. Indeed, we hope that the main takeaway from our submission is that optimistic exploration in imitation learning is beneficial. This fact does not seem to be very well known in the current IL literature.
>
> ***Very recently, Moulin, … manuscript.***
>
> Thanks to the pointer to this reference we missed. We will add it in our revision
>
> ***The main contribution … theme).***
>
> In our opinion, the 2 contributions nicely complement each other. Indeed, our goal was to find a method achieving these two desiderata simultaneously:
>
> 1) Guarantees in the tabular case.
> 2) Convincing empirical performance in continuous control experiments with minimum algorithm modifications ( that is the way the critics are trained).
>
> The ensemble based exploration technique implemented in SOAR is one possible way to achieve this goal.
> Notice that, the tabular guarantees per se could also have been achieved with classic analysis leveraging exploration bonuses but it is very unclear how bonus based algorithms can be implemented with neural networks.
>
>
> ***In Algorithm 1, line 11, it seems like $k$
>  should rather be  $K$***
>
> Thanks for catching the typo ! We will fix it.
>
> ***Line 372 refers to an "attainable range". How is this defined?***
>
> We will clarify this. By attainable range we mean the interval in which the state value function can take values. Since for all $s \in \mathcal{S}$, we have that $V(s) \in [0, (1-\gamma)^{-1}]$ we consider as attainable range the interval $[0, (1-\gamma)^{-1}]$.
>
> Thanks again for the time spent reviewing our paper, we remain available for further clarification if needed.
>
> Best,
> Authors

---

### Official Review · Reviewer_EDMt · 2025-03-13

**Overall Recommendation:** 4

**Summary:**

This paper presents SOAR: an algorithmic template that learns a policy from expert demonstrations with a primal dual style algorithm that alternates cost and policy updates. The method boosts consistently the performance of imitation learning algorithms based on Soft Actor Critic across various Mujoco tasks.

## update after rebuttal
I confirm my score. Authors addressed comments and added clarity and results to the original submission.

**Claims And Evidence:**

Claims are presented clearly and supported by clear and convincing evidence including proofs and sketch proofs.

**Essential References Not Discussed:**

N/A

**Experimental Designs Or Analyses:**

The experiments conducted replicate sets of experiments that are common and well known, as well as using well know algorithms for comparison. The design of the experiments and their analysis is valid and supports the claims.

**Methods And Evaluation Criteria:**

The proposed method and evaluation seem reasonable and informative for the application at hand. Comparisons across variants and different algorithms also support the claims made.

**Other Comments Or Suggestions:**

Please capitalize algorithms names in your figures.

**Other Strengths And Weaknesses:**

The paper is well presented and significant. Claims, theoretical contributions and experiments are presented in a clear way. It can have impact on imitation learning as well as applications such as in robotics, where reducing exploration time is a significant issue.

**Questions For Authors:**

Please refer to the other comments.

**Relation To Broader Scientific Literature:**

This work is relevant for the imitation learning community, and major parts of the literature therein are covered.

**Theoretical Claims:**

I reviewed the theoretical claims and equations, although I did not verify all mathematical derivations in detail. While it is possible that I may have overlooked some aspects, the theoretical claims appear to be correct to the best of my understanding.

---

> ### Author Rebuttal · Authors · 2025-03-27
>
> Dear reviewer,
>
> Thanks a lot for appreciating our work !
>
> We will make sure to capitalize the algorithm names in the figures.
>
> Thanks again for the time spent reviewing our paper, we remain available for further clarification if needed.
>
> Best,
> Authors

---

> > ### Comment · Reviewer_EDMt · 2025-04-02
> >
> > Thank you for addressing my and other reviewers' comments.

---

### Official Review · Reviewer_Kkc5 · 2025-03-16

**Overall Recommendation:** 3

**Summary:**

The authors propose an optimistic $Q$-function update for SAC-based inverse RL algorithms. They prove sample complexity benefits in the tabular setting and show an approximation of their idealized algorithm can improve the performance of a suite of prior approaches with minimal increase in complexity.

## Update After Rebuttal
The authors added in a discussion of some of the points of confusion I originally had to the paper and added in experiments I specifically requested. I already factored them doing so into my evaluation of the paper, and hence maintain my score

**Claims And Evidence:**

Yes.

**Essential References Not Discussed:**

I would make sure to cite https://papers.nips.cc/paper_files/paper/2007/hash/ca3ec598002d2e7662e2ef4bdd58278b-Abstract.html as the first paper to introduce the no-regret vs. no-regret flavor of inverse RL. It would also be good to add more of a discussion of https://arxiv.org/abs/2205.15397 as they also use several variants of deep ensembles in imitation learning -- some of the aggregation methods they explore might be interesting for you. Other SAC-based IRL algorithms that would be good to cite / potentially try out SOAR on include AdRIL (https://arxiv.org/abs/2103.03236), HyPE (https://arxiv.org/abs/2402.08848), and SMILING (https://arxiv.org/abs/2410.13855) -- the first and third have fairly unique ways of doing the cost function update, while the second has some interesting ideas re: policy updates. You should also cite https://www.sciencedirect.com/science/article/pii/S002200009791504X to be more transparent about your game-solving strategy for Eq. 1.

**Experimental Designs Or Analyses:**

Please include standard error bars in all the plots and specify the number of seeds used to compute the results in Figure 1.

**Methods And Evaluation Criteria:**

Yes.

**Other Comments Or Suggestions:**

There are several parts of this paper where I think the writing could be improved to make things more accessible:

- Define $\epsilon$ in the abstract or postpone its usage to later. Also, I think you meant to say "tied" not "tight" in the second paragraph.

- From the way the paper is written, it is not at all clear that you could apply this technique to *all* SAC-based IRL methods. I'd suggest heavily emphasizing this in the abstract / intro to properly convey the general utility of the techniques you developed.

- Treating the value function as a vector and the performance of a policy as an inner product (Line 80)  likely going to confuse a lot of people -- speaking from experience here.

- I think the algorithm writeup is fairly sloppy and hard to follow if you don't already have a strong background in this sort of thing. I might suggest presenting the theory independently first before broadening out and discussing, e.g., the different ways of learning cost functions in practice.

- Section 3.1 mostly reads as a statement of facts rather than a coherent narrative / clear proof sketch -- I strongly recommend rewriting this section to make it easier to follow if you aren't already an expert. I would also cut some of the alternatives (e.g. state-only, the mean - std. ideas) and postpone them to the appendix.

- It would also be good to be more transparent about what you're doing in your description of the algorithms (e.g. I'd explicitly say you're doing NPG in Line 14 of Algorithm 2 and Zinkevich-style OGD in Line 8 of Algorithm 3). It then becomes much more clear that you're running two no-regret players against eachother to compute equilibria -- I would explicitly call this out, perhaps citing the list of "primal" algorithms in https://arxiv.org/abs/2103.03236 that follow a similar strategy.

- In Line 250, you could perhaps cite https://dl.acm.org/doi/abs/10.1145/1390156.1390286 as an example of how if you knew the dynamics in closed form, you could collapse the trajectory-level mixture to a single policy.

**Other Strengths And Weaknesses:**

I think the greatest strength of this paper is the broadness of the technique proposed. I think the greatest weakness (which can easily be fixed) is the clarity with which it is presented.

**Questions For Authors:**

1. Have you thought about whether your ideas could be useful for standard RL problems where we know the reward function? Nothing here seems tied to IRL per se in that portion of your analysis. It might be interesting to try out the optimistic critic on a variety of RL problems and potentially have another contribution with minimal theoretical overhead.

2. Why are you adding 2 and not 1 to the denominator of the fraction in Line 11 of Algo 2?

3. Given you're running two no-regret players against each other to solve the game, can you get last iterate convergence via optimism (in the other, game-theoretic sense :p)? If that works, it might be a nice result to add in. If not, I would still think explaining where the proof breaks to be interesting.

4. If it's computationally feasible, could you try the Bayesian dropout version of an ensemble (https://proceedings.mlr.press/v48/gal16.pdf) and report the performance? This way, SOAR has no memory overhead over standard SAC.

5. In Remark 3.1, when you say $L=1$ for most SAC implementations used for IRL, do you mean they have a single *pair* of critics? All SAC implementations I'm aware of don't have a single critic -- would be good to clarify this.

**Relation To Broader Scientific Literature:**

SOAR could realistically improve a wide set of prior SAC-based IRL methods, which I find quite interesting!

**Theoretical Claims:**

I read all the theorems and skimmed the proofs -- both looked like what I expected to see.

---

> ### Author Rebuttal · Authors · 2025-03-27
>
> Dear reviewer,
>
> thanks a lot for appreciating our work
> and for all the suggestions ! Please find below the responses to your questions.
>
> ***Have you thought ... overhead.***
>  We agree with the reviewer! We think that interesting next steps are applying the exploration technique based on multiple critics in RL and RLHF. For this paper, we feel that it would be better to keep the message more coincise to imitation learning. We can definitely mention in the future directions this point.
>
> ***Why are you adding 2 ...***
> A larger additional weight in the denominator is needed for technical reasons.
> In particular, to invoke Cassel et al. (2024, Lemma 3) in the proof of Lemma B.1 which in turns applies Corollary 1 (b) in [1]. Intuitively, adding $2$ rather than $1$ in the denominator is needed to ensure that the estimators are more optimistically biased for a small number of visits.
>
> [1] Wiklund 2023,  Another look at binomial and related distributions exceeding values close to their centre
>
> ***Given you're running ... interesting.***
>
> Thanks for the suggestion. It would be nice to see if optimistic gradient descent + optimistic exploration comes with any benefit.
>
> We speculate that proving last iterate convergence using optimistic updates for both reward and cost should be possible but it could deteriorate the sample complexity.
> An alternative approach to get last iterate could be to strongly regularize the problem as in [2] but this comes at the cost of worst rates in $K$.
>
> In the attempt we did to prove last iterate convergence with optimistic gradient descent ascent the problem is controlling the expected bonus under the last iterate distribution. Indeed, the technique we use to bound the average bonuses in Lemma C.2 does not seem helpful towards this goal.
>
> [2] Müller et al. Truly No-Regret Learning in Constrained MDPs
>
> ***If it's ... SOAR has no memory overhead over standard SAC.***
>
> Thanks for the suggestion ! We think that at the practical level this uncertainty quantification technique can be used effectively.
> We are more hesitant in saying whether guarantees can be proven about this technique. We will definitely add a discussion about this technique.
>
> ***In Remark 3.1, when you ... clarify this.***
>
> The reviewer is right. We meant a single pair of critics in Remark 3.1 and we will clarify this in our revision. Thanks a lot for noticing this !
>
> ***Response to the section "Other comments and suggestions"***
>
> Thanks for pointing out the missing references, for suggesting how to improve the abstract and the overall writing (especially for Section 3.1 and for the presentation of the algorithm) ! We are happy to implement these changes and include a discussion about the missing references in our revision.
>
> Finally, thanks for suggesting AdRIL, HyPe and SMILING as possible algorithms to be boosted with SOAR. During the rebuttal time,
> we run a SOAR boosted version of HyPe and we noticed that it improves over the base version of Hype in Ant, Humanoid and Hopper. The results are available here https://imgur.com/a/2Zse8xl
>
>
> Thanks again for the time spent reviewing our paper, we remain available for further clarification if needed.
>
> Best,
> Authors

---

### Official Review · Reviewer_PNta · 2025-03-21

**Overall Recommendation:** 3

**Summary:**

This paper introduces IL-SOAR, an imitation learning (IL) framework designed to bridge the gap between theoretical sample efficiency and practical scalability. The IL-SOAR framework alternates between updating a cost function (critic) and updating the policy (actor).

A key contribution of IL-SOAR is the introduction of an optimistic critic, leveraging multiple critic networks to quantify uncertainty. Specifically, IL-SOAR constructs an optimistic critic that systematically underestimates the true expected cost, thus effectively promoting exploration.

The authors provide theoretical guarantees for the efficiency of this optimistic critic approach through regret analysis in a tabular setting, explicitly analyzing two aggregation methods for optimism: Min and Mean-Std.

Practically, IL-SOAR extends this principle of optimism to continuous domains by applying the optimistic critic to pre-existing approaches, such as CISL, ML-IRL. Empirical results demonstrate superior performance relative to standard approaches, highlighting IL-SOAR as a theoretically grounded and practically effective online imitation learning method.

**Claims And Evidence:**

**Theoretical Claim:**

IL-SOAR provides a theoretically efficient imitation learning framework.

**Evidence:**

The authors present a detailed regret analysis for the tabular setting, considering both min and mean-std aggregations. They derive explicit theoretical guarantees, demonstrating that IL-SOAR achieves a regret bound matching established results in terms of $\epsilon$, the value error between the imitator and the expert. This analysis rigorously supports the theoretical efficiency claim regarding both the number of critics and the number of samples.

**Experimental Claim:**

IL-SOAR demonstrates practical scalability and effectively enhances vanilla methods (which utilize a single critic) in continuous state-action imitation learning tasks under both state-only and state-action settings.

**Evidence:**

The authors demonstrate empirical results through experiments conducted on several MuJoCo continuous control environments (Ant, Hopper, Walker2d, and Humanoid). They compare IL-SOAR methods (ML-IRL+SOAR, RKL+SOAR, CSIL+SOAR) against their vanilla methods (ML-IRL, RKL, CSIL) and strong baselines (GAIL, SQIL, OPT-AIL), showing consistent and significant improvements across different algorithms and tasks. These empirical results strongly support the practical effectiveness claim.

**Essential References Not Discussed:**

I believe the manuscript includes all essential related references.

**Experimental Designs Or Analyses:**

Experimental design seems quite valid. About analyses on this, I remain some comments on a further section.

**Methods And Evaluation Criteria:**

### Methods

The IL-SOAR method employs a simple structure commonly used in inverse reinforcement learning (IRL), alternating between updating the cost function (critic) and updating the policy (actor). A key methodological innovation is the optimistic critic, constructed using multiple critic networks to estimate uncertainty. This critic systematically underestimates the true expected cost, thereby encouraging effective exploration.

The authors evaluate two explicit methods for aggregating critic outputs to achieve optimism:

1. Min aggregation (selecting the minimum critic estimate) (for a tabular case)
2. Mean-Std aggregation (using the mean estimate minus the standard deviation across critics) (for both tabular and continuous cases)

### Evaluation Criteria

The proposed method is empirically evaluated based on episode returns in standard continuous control benchmarks (MuJoCo environments such as Ant, Hopper, Walker2d, and Humanoid). These environments are widely recognized for assessing reinforcement and imitation learning methods, making them appropriate for evaluating practical performance.

However, it is unclear whether these benchmarks genuinely require exploration to successfully recover the expert policy. To better highlight the importance of exploration, additional experiments using explicitly exploration-critical tasks would strengthen the central claims of the paper.

**Other Comments Or Suggestions:**

Typos in the legends of Figures 3 and 7, as well as in Section D, should be corrected: “cisl” should be revised to “csil” (Coherent Soft Imitation Learning).

**Other Strengths And Weaknesses:**

### Strengths

- The paper effectively bridges theoretical principle with practical applicability, providing both solid theoretical guarantees and robust empirical results.
- Experimental results demonstrates significant improvements over vanilla imitation learning approaches.
- The method is generalizable and easily integrable into existing IL frameworks, providing broader applicability within the imitation learning community.

### Weaknesses

- Although the paper emphasizes exploration in IL, the evaluation benchmarks presented are not particularly exploration-critical tasks, potentially limiting insights into the effectiveness of the proposed exploration strategy in imitation learning. Including experimental results on explicitly exploration-critical benchmarks would significantly enhance the paper’s impact.
- There appears to be a gap between theoretical expectations and practical observations regarding the relationship between performance and the number of critics. Specifically, Theorem 3.4 indicates that the upper bound of the average regret decreases as the number of critics (K) increases. However, Figure 4, assuming the number of neural networks corresponds to the number of Q-value functions, shows that performance improvement is not directly proportional to an increase in the number of Q-value functions. This finding is counterintuitive, as increased uncertainty estimation accuracy from more Q-value functions might be expected to lead to better performance. Conversely, in a pessimistic approach [1] within the offline RL context, performance consistently and significantly improves as the number of Q-value functions increases, highlighting a notable contrast to the current observation.

[1] An et al., “Uncertainty-Based Offline Reinforcement Learning with Diversified Q-Ensemble”, NeurIPS 2021.

**Questions For Authors:**

**Question 1.** This paper proposes two aggregation methods for Q-values in tabular settings: Min and Mean-Std. Under what conditions or scenarios would each aggregation method be advantageous compared to the other?

Additionally, why is the mean-std aggregation method particularly considered for OptimisticQ-NN in continuous state-action scenarios, rather than the min aggregation method?

**Question 2.** Why is the performance improvement not directly proportional to the increase in the number of Q-value functions, as shown in Figure 4? Did the authors try the experiment with a more larger scale? (approximately 10~100 Q-value functions)

**Relation To Broader Scientific Literature:**

The key contributions of this paper are summarized as follows:

1.	A theoretical analysis demonstrating that optimism in the face of uncertainty can lead to an efficient imitation learning algorithm.
2.	The introduction of the IL-SOAR framework, which alternates between multiple critic (cost) updates and actor updates using an optimistic critic approach.
3.	Empirical results showing that IL-SOAR improves imitation performance in continuous control domains.

To the best of my knowledge, these contributions are novel within the imitation learning community.

**Theoretical Claims:**

Although I have not verified every detail of the theoretical claims and proofs, they appear rigorous and well-supported, with comprehensive derivations provided in the appendix.

---

> ### Author Rebuttal · Authors · 2025-03-27
>
> Dear reviewer,
>
> Thanks a lot for appreciating our work and your careful review!
>
> Please find the answers to your questions/remarks in the following.
>
> ***..experiments using explicitly exploration-critical tasks would strengthen the central claims of the paper.***
>
> In order to assess the exploration capability of our algorithm we consider state only imitation learning in an
> hard tabular environment which is usually used to demonstrate lower bounds.
> We refer to [1] section 4 for a detailed description of the environment. We make it slightly harder considering 20 actions (instead of 2 actions as in [1]). Among the actions available to the learner there is one action $a^\star$ that has slightly higher probability to maintain the learner in the state visited more often by the expert.
> Since, we are in imitation learning from states only setting, the learner can not observe such action $a^\star$ therefore it has to explore efficiently the action set to estimate $a^\star$.
>
> We use this environment for this experiment https://imgur.com/a/HI3LOeq. The results show that exploration is necessary in this environment. Indeed, it is evident that for $L=1$, that is when the algorithm does not have an  exploration mechanism, the learner fails at achieving expert performance while the learner succedes when multiple critics are added.  ( We will comment on the choice of $L$ later in our response)
>
> Regarding continuous control experiments, we are not really sure about which environment could be used to assess the exploration capability of an algorithm. Does the reviewer have some environments in mind ?
>
> [1] Osband & Van Roy 2016, On Lower Bounds for Regret in Reinforcement Learning
>
> ***There appears to be a gap between theoretical expectations and practical observations...***
>
> In figure 4 increasing the number of critics does not improve the performance further but this is not in contradiction with the theory.
> Indeed, notice that in Theorem 3.4 $K$ denotes the number of iterations while the number of critics is denoted by $L$. The performance of the algorithm is not expected to improve in a monotone way at the increase of $L$ because of the following tradeoff:
>
> 1) $L$ should be large enough to ensure high enough probability of optimism according to Corollary 4.10 ( see also Lemma B.1)
> 2) $L$ should be small enough to guarantee a tight bound on $\delta^k$ according to Lemma 4.12 where the number of critics L appears at the numerator.
>
> Therefore, it is expected from the theoretical study that an excessively large value of $L$ leads to worst sample complexity guarantees. Ideally, $L$ should be set equal to the smallest value guaranteeing optimism with probability $1-\delta$. This is the reasoning behind setting $L = 7/2\log(2 SAK / \delta)$ in our paper.
> In order to verify the theory, we add an ablation on the number of $Q$ estimators ( going up to 100 as suggested) in the tabular setting which verifies that there is indeed a tradeoff in the choice of $L$. The image is visible via this anonymous link https://imgur.com/a/HI3LOeq
>
> ***Question 1.***
> From the theoretical analysis presented here the two aggregation rules are equivalent. In practice we think that mean+std offers more flexibility in tuning the clipping parameter. Therefore, while equivalent in the worst case the std based aggregation rule seems to achieve better instance dependent results. It would be interesting to confirm this observation theoretically in future work.
>
> ***Question 2.***
>
> We did not try to increase the number of critics above 10 in the continuous states and actions experiments with neural networks. We think this should make the algorithm’s computational cost excessively high compared to a single critic implementation. In the tabular environment considered here https://imgur.com/a/HI3LOeq we try a larger number of critics and see that there is a tradeoff in the choice of $L$ ( please see explanation above).
>
> Thanks again for the time spent reviewing our paper, we remain available for further clarification if needed.
>
> Best,
> Authors

---

### Decision · Program_Chairs · 2025-05-01

**Decision:**

Accept (poster)

**Comment:**

This paper presents a new algorithm to learn a policy from expert demonstrations by alternating between cost and policy updates. This method uses an optimistic -function update for SAC-based inverse RL algorithm. The sample efficiency of the proposed method is proven theoretically. The efficiency is also proven practically on MuJoCo environments, where this method is shown to outperform certain alternatives.
The paper is well-written, and the combination of a strong theoretical analysis and a sound empirical evaluation makes this paper well-suited for ICML.